# Polymersomes preventing brain infiltration of CD177+ neutrophils to mitigate hemorrhagic transformation post-tPA thrombolysis

Zhenhua Wang[1], Hua Liu[2], Zhiyao Xu[2], Cheng Huang[1], Xing Guo [1] ✉ & Shaobing Zhou [1]

As an intravenous thrombolytic agent, tissue plasminogen activator (tPA) is limited by hemorrhagic transformation (HT) and a narrow therapeutic time window. Here we develop ROS-triggered polymersomes conjugated with fibrin-targeting CREKA peptide (CP) for tPA encapsulation. CP@tPA achieves efficient thrombosis targeting and enhanced thrombolytic efficacy, however, it has not been able to avert the occurrence of hemorrhage subsequent to thrombolysis. Building from our clinical discovery that stroke patients suffer from tPA-induced HT featured strong expression of *CD177*, recombinant CD177 protein (rCD177) was loaded into CP polymersomes (CP@rCD177) as nanomedicine and administrated prior to CP@tPA thrombolysis. rCD177 can be released in response to elevated ROS within the obstructed vessels, which binds to endothelial cells through interacting with CD31 and efficiently prevents the migration of CD177+ neutrophils into the brain parenchyma. The reduced CD177+ neutrophils suppress the generation of neutrophil extracellular traps (NETs), thereby dampening the inflammatory activation of microglia and ultimately improving the prognosis of HT. This innovative strategy presents a promising avenue for attenuating hemorrhagic complications post-thrombolysis.

Ischemic stroke, a leading cause of long-term disability and death globally, is characterized by the occlusion of cerebral blood vessels[1–3]. The rapid restoration of blood flow through thrombolytic therapy is crucial for salvaging ischemic penumbra and minimizing neurological deficits. Tissue plasminogen activator (tPA) has been the only U.S. Food and Drug Administration (FDA)-approved thrombolytic drug for the treatment of acute ischemic stroke[4,5]. By intravenous injection, tPA catalyzes the conversion of plasminogen to plasmin, which in turn breaks down the fibrin mesh of the clot, thereby restoring blood flow to the ischemic area[6]. Despite its proven efficacy in thrombolysis, tPA therapy still remains limitations. Its short half-life necessitates

continuous infusion[7], and there is a potential risk of hemorrhagic transformation (HT)[8,9], which can exacerbate patient outcomes and augment mortality rates. This risk necessitates stringent adherence to a 4.5-h time window from symptom onset.

The mechanism underlying tPA-induced HT is complex and multifactorial, involving the disruption of the blood-brain barrier (BBB), inflammatory responses, and oxidative stress[10,11]. BBB plays a critical role in maintaining the homeostasis of the central nervous system, and its integrity is compromised following ischemic stroke, particularly after reperfusion with tPA[12]. The disruption of the BBB allows for the extravasation of blood into the brain parenchyma to cause HT.

[1]Institute of Biomedical Engineering, College of Medicine, Southwest Jiaotong University, Chengdu, PR China. [2]Department of Neurology, The Third People's Hospital of Chengdu, Chengdu, PR China. ✉e-mail: xingguo@swjtu.edu.cn

Additionally, inflammatory processes are also implicated in tPA-related HT. Post-ischemic inflammation triggered by tPA, involves the recruitment and activation of neutrophils and other immune cells, which can exacerbate brain injury and increase the risk of HT[13–15]. Reactive oxygen species (ROS) generated during reperfusion can further amplify the inflammatory response and contribute to neuronal damage and BBB dysfunction[16,17]. Understanding the intricate mechanisms of tPA-induced HT is crucial for developing strategies to mitigate this complication.

Our clinical observations have uncovered a significant finding in the context of HT patients who have undergone tPA therapy. We have noted a robust expression of *CD177* in their peripheral blood. As a cell surface glycoprotein mainly expressed on neutrophils, CD177 engages in a critical interaction with platelet endothelial cell adhesion molecule-1 (PECAM-1, also known as CD31)[18,19]. This interaction is instrumental in modulating the phosphorylation of CD31 and the integrity of endothelial junctions, which in turn facilitates the

transendothelial migration of neutrophils[19–21]. Moreover, CD177+ neutrophils are capable of generating elevated levels of ROS, neutrophil extracellular traps (NETs), and myeloperoxidase (MPO), molecules that are known to disrupt the BBB and contribute to HT[22–24]. This suggests that CD177+ neutrophils serve as critical predictive biomarkers for thrombolysis-induced hemorrhages, thereby emphasizing their potential as therapeutic targets to mitigate the risk of HT.

In this study, we develop thrombus-targeting and ROS-triggered polymersomes to separately load tPA and recombinant CD177 protein (rCD177) for thrombolysis while concurrently inhibiting HT (Fig. 1). Given the abundance of fibrin on the surface of both patient and mouse blood clots (Supplementary Fig. 1), fibrin-targeting peptide (Cys-Arg-Glu-Lys-Ala, CREKA) was anchored onto the surface of polymersomes to specifically recognize fibrin[3,25]. Additionally, phenylboronic acid (PBA) was incorporated into polyethylene glycol (PEG) and polylactic acid (PLA) as ROS-responsive triggers to facilitate the structural disintegration of the polymersomes at the thrombus site[26,27].

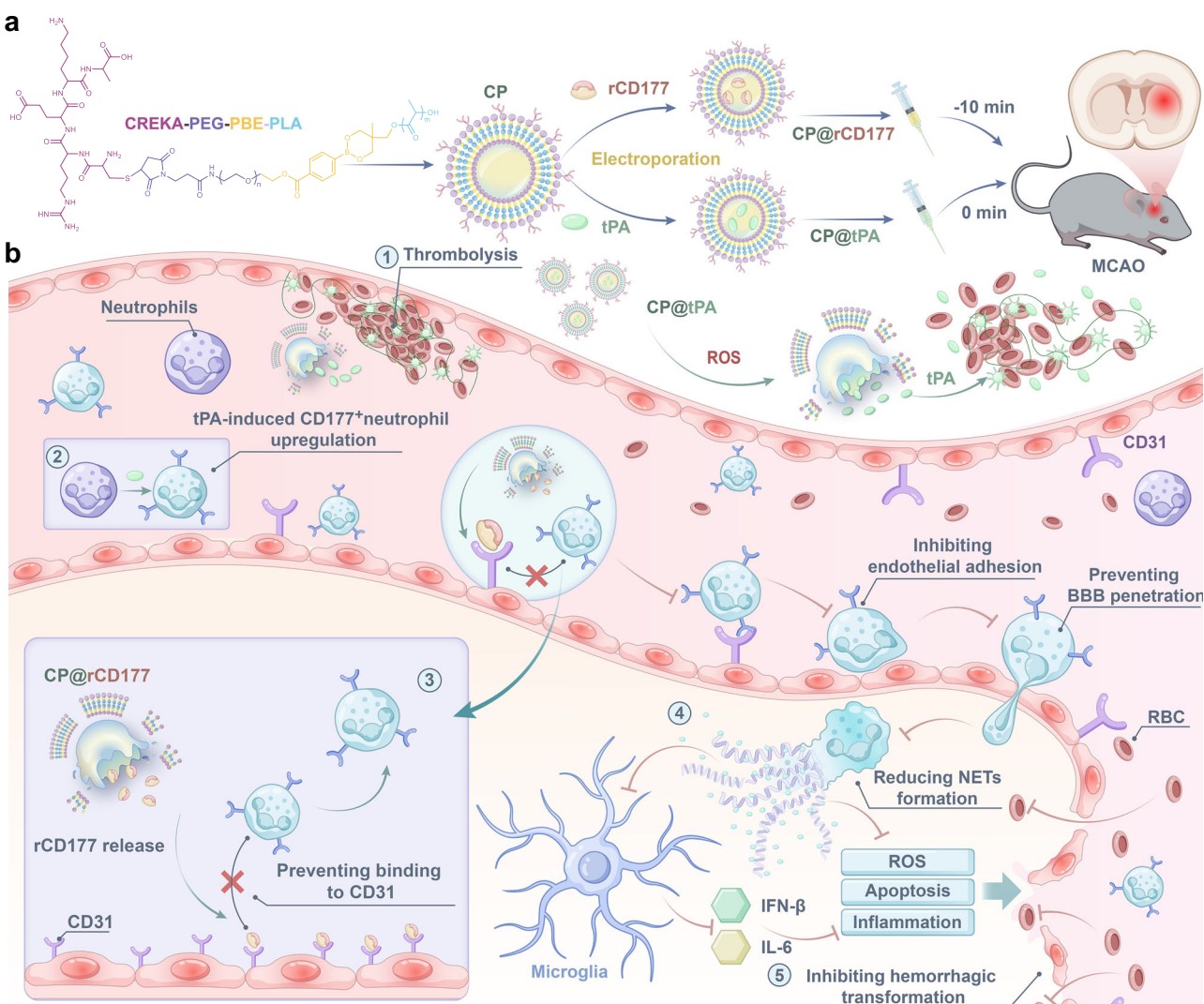

**Fig. 1 | Schematic illustration of the polymersome preparation and the mechanism by which polymersomes inhibit hemorrhagic transformation following tPA thrombolysis is depicted. a** CP polymersomes were fabricated through a process involving the conjugation of the CREKA peptide to PEG and the incorporation of PBA into PEG and PLA. Subsequently, tPA and rCD177 were encapsulated within the CP polymersomes via electroporation. **b** After intravenous injection into MCAO mice, CP polymersomes could specifically target and accumulate at the thrombus site by recognizing fibrin through the CREKA peptide.

Subsequently, the polymersomes would disintegrate upon ROS trigger, releasing tPA to facilitate thrombolysis and restore blood flow. Moreover, the pre-treatment with CD@rCD177 before thrombolysis could effectively block the interaction between tPA-upregulated CD177+ neutrophils and endothelial cells, reducing the brain infiltration of CD177+ neutrophils and decreasing the formation of NETs. Ultimately, this cascade of events would alleviate inflammation and apoptosis, inhibit angiogenesis, and thus achieve the goal of suppressing hemorrhagic transformation.

This process leads to a targeted release of tPA, effectively dissolving the thrombus. Additionally, the polymersomes can prolong the half-life of tPA, thereby achieving sustained thrombolytic effects with a single administration. Intervention with CP@rCD177 prior to CP@tPA thrombolysis can impede the brain infiltration of CD177[+] neutrophils by binding to CD31, diminishing the generation of NETs and consequently curbing the inflammatory activation of microglia. Notably, the co-administration of CP@rCD177 with CP@tPA significantly preserve the integrity of the BBB and mitigate the incidence of tPA-associated hemorrhage, thereby boosting its therapeutic efficacy in ischemic stroke. Our findings confirm the role of CD177[+] neutrophils in hemorrhage following thrombolysis and offer a novel therapeutic approach for the clinical attenuation of tPA-induced HT.

## Results

### Preparation and characterization of the polymersomes

To prepare CP polymers, PEG was connected with PBA and subsequently initiated the ring-opening polymerization of D,L-LA to get PEPLA, which was conjugated with CREKA-peptide to produce the final product (Supplementary Fig. 2). The chemical structures of CP and the intermediates were confirmed by $^1$H nuclear magnetic resonance ($^1$H NMR) (Supplementary Fig. 3). The CP polymersomes were prepared by solvent evaporation and tPA was encapsulated within the vesicles by electroporation (Fig. 2a). As controls, PEG-PLA (PELA) and PEG-PBE-PLA (PEPLA) were loaded with tPA to prepare Null@tPA and P@tPA, respectively. Dynamic light scattering (DLS) measurements indicated that the particle size and zeta potential of CP@tPA are 125.7 nm and $-3.43 \pm 0.74$ mV, respectively (Supplementary Table 1). The drug loading content and encapsulation efficiency of CP@tPA are 7.55% and 81.73%, respectively.

The phenylboronic moiety undergoes sequential oxidation and hydrolysis when stimulated by $H_2O_2$[25,28]. To confirm the ROS-triggered disruption of boronic ester bonds in polymersomes (Fig. 2a), transmission electron microscopy (TEM) was performed to examine the morphology changes in Null@tPA, P@tPA, and CP@tPA. Compared to the polymersomes that maintain a closed vesicular morphology with a bilayer structure under normal condition, significant structural damage was observed in the boronic ester-modified P@tPA and CP@tPA after incubation with $H_2O_2$ (Fig. 2b). Consistent with the TEM images, DLS showed size change in the ROS (reactive oxygen species)-responsive polymersomes upon $H_2O_2$ exposure (Fig. 2c and Supplementary Fig. 4a). In contrast, the structure and particle size of the non-responsive Null@tPA remained unchanged (Fig. 2b and Supplementary Fig. 5a). The ROS-induced destruction of the polymersome structure inevitably accelerates drug release, resulting in a 2.68-fold and 2.3-fold increase in tPA release from P@tPA and CP@tPA after 24 h of incubation with $H_2O_2$ (Fig. 2d and Supplementary Fig. 4b). No significant change in tPA release was observed when Null were treated with $H_2O_2$ (Supplementary Fig. 5b). Additionally, all three polymersomes exhibited good size stability in phosphate buffered saline (PBS), with slight size changes over a week (Fig. 2e). To ensure in vivo applicability of CP@tPA, we further evaluated their colloidal stability in PBS and long-term stability. The results indicated that CP@tPA exhibited excellent stability within 8 h in PBS (Supplementary Fig. 6a), and maintained structural integrity for up to 30 days when stored at 4 °C in the dark (Supplementary Fig. 6b). The ROS scavenging ability of Null is negligible compared to that of P and CP, whereas the responsive polymersomes reduced $H_2O_2$ content by more than 80% (Fig. 2f). Despite the widespread clinical use of tPA as a thrombolytic agent, its therapeutic application is constrained by the rapid blood clearance due to its short half-life[29]. The in vitro cytotoxicity indicated that CP@tPA possessed good biocompatibility when incubated with bEnd.3 cells for 24 h, showing cell

vitality over 85% (Supplementary Fig. 7). The hemolysis rate of CP@tPA was below 5%, and the blood clotting index (BCI) was comparable to that of the saline group (Supplementary Fig. 8), demonstrating good blood compatibility. To investigate whether the polymersome encapsulation can prolong the blood circulation of tPA, the pharmacokinetics of CP@tPA was evaluated. We found that CP polymersomes significantly extended the half-life of tPA from 6.99 min to 65.75 min (Supplementary Fig. 9 and Supplementary Table 2). At 2 h post-administration, CP@tPA maintained 31.79% of its initial tPA content.

### CP facilitates thrombus targeting

We first explored the binding affinity of CREKA-modified polymersomes to fibrin using in vitro clots. After treating the thrombi with ICG-loaded polymersomes, CP@ICG group displayed remarkably stronger fluorescence than Null and P groups at each time point (Supplementary Fig. 10a), indicating enhanced targeting property mediated by CREKA peptide. Specifically, the fluorescence intensity of the CP@ICG group was 2.03 times and 2.06 times higher than that of the Null@ICG and P@ICG groups, respectively (Supplementary Fig. 10b). Additionally, the distribution of polymersomes on the thrombus was observed through scanning electron microscope (SEM). The images revealed that CP were abundantly anchored to fibrin on the thrombus surface, whereas significantly fewer thrombus adhesions were observed in Null and P groups (Fig. 2g).

The thrombus-targeting capability of the polymersomes was subsequently assessed in stroke mice. As shown in Fig. 2h, CP@ICG exhibited a significantly stronger fluorescence signal in the thrombus-occluded area compared to the non-targeted groups. Quantitative analysis of the average radiance efficiency at 2 h post-injection indicated that the fluorescence intensity of CP@ICG in the infarct brain was 5.14-fold and 4.9-fold higher than that of the Null@ICG and P@ICG groups, respectively (Supplementary Fig. 11). Ex vivo fluorescence analysis further confirmed that CP@ICG possessed a strong signal in the ischemic area (Supplementary Fig. 12a). Specifically, the CP@ICG intensity in the infarct brain was 2.7-fold and 3.77-fold higher than that in the Null@ICG and P@ICG groups, respectively (Supplementary Fig. 12b). Furthermore, immunofluorescence staining with antifibrin antibody demonstrated that CP@ICG-treated brain tissues exhibited a remarkably high degree of ICG signal colocalization with fibrin (Fig. 2i), indicating specific targeting to fibrin-rich thrombi. These results proved that CP polymersomes could effectively accumulated at the thrombus site, thereby enhancing tPA concentration at the lesion site and contributing to thrombolytic effect. Histopathological and serum biochemical analyses revealed no significant damage to major organs or impairment of hepatic and renal functions following CP@tPA treatment (Supplementary Fig. 13a and 13b), suggesting good biocompatibility and biosafety of these nanodrugs in vivo.

### CP@tPA promotes thrombolysis

CP@tPA exerts thrombolytic activity and restores blood flow through CREKA-mediated thrombus targeting and ROS-triggered payload release (Fig. 3a). We systematically evaluated its thrombolytic performance across three distinct thrombus models, including carotid artery thrombus, lower extremity arterial thrombus, and middle cerebral artery occlusion (MCAO). In the lower extremity arterial thrombus model, free tPA failed to achieve complete clot dissolution within 2 h (Fig. 3b). In contrast, CP@tPA demonstrated progressive thrombus resolution, evidenced by gradual clot lightening and complete restoration of blood flow within the same timeframe. Thrombus-induced vascular obstruction creates a hypoxic environment, promoting ROS production and leading to platelet activation and aggregation[30,31]. Using two-photon microscopy, we observed persistent platelet aggregation in the lower extremity arterial thrombosis, which exacerbates vascular occlusion (Fig. 3c). CP@tPA treatment

accelerated platelet signal reduction (vs. free tPA) and restored functional perfusion after 1 h. Mechanistically, CP@tPA attenuated thrombogenic inflammation by reducing soluble CD40 ligand

(sCD40L) levels – critical mediators of endothelial activation[32,33]. The plasma levels of sCD40L in the CP@tPA group were reduced by 41.08% and 33.58% compared to the saline group and tPA group, respectively

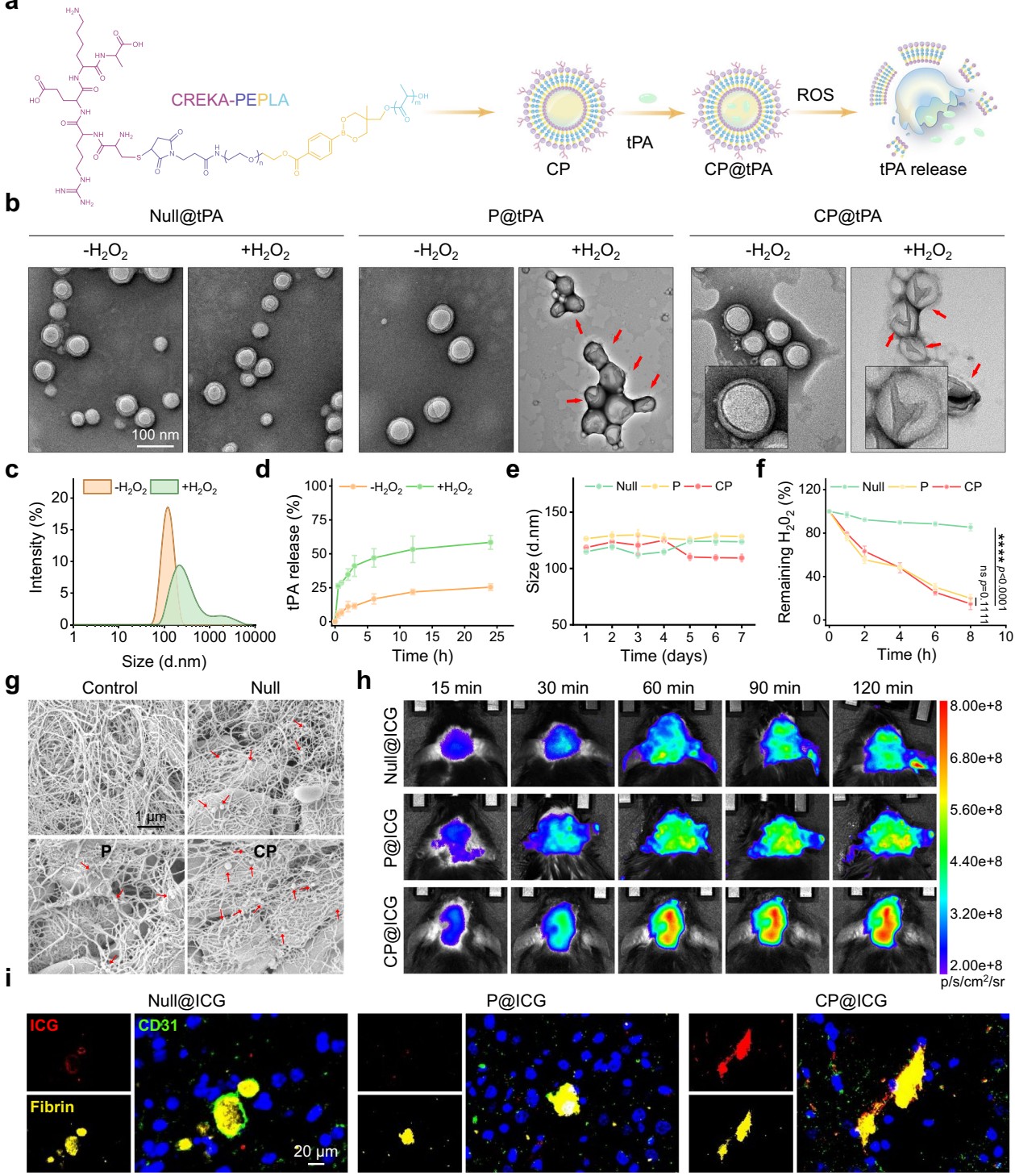

**Fig. 2 | Physiochemical properties and thrombosis targeting of polymersomes.**
**a** Schematic illustration of CP@tPA polymersomes preparation and disintegration upon ROS trigger. **b** Representative TEM images of polymersomes in PBS or $H_2O_2$. The disintegrated vesicles are pointed by red arrows. **c** Size distribution of CP polymersomes in PBS or $H_2O_2$. **d** In vitro tPA release from CP polymersomes in PBS or $H_2O_2$ ($n = 3$ biologically independent samples). **e** Size stability of polymersomes in PBS within 7 days ($n = 3$ biologically independent samples). **f** ROS removal capability of different polymersomes ($n = 3$ biologically independent samples). **g** SEM images showing thrombus targeting of polymersomes. The vesicles adhered to thrombus are pointed by red arrows. **h** In vivo fluorescence imaging of ICG-loaded polymersomes in the brains of MCAO mice. **i** Immunofluorescence imaging of ICG-loaded polymersomes (red) and fibrin-labeled thrombus (yellow) in the infarct brain 2 h post-injection. Blood vessels were labeled with CD31 (green) and nucleus were stained with DAPI (blue). The results in (**b, g, i**) were representative of three independent experiments. Data in (**d−f**) were presented as mean ± SD. *P* values were calculated by two-way followed by Bonferroni's post-hoc test. Source data are provided as a Source Data file.

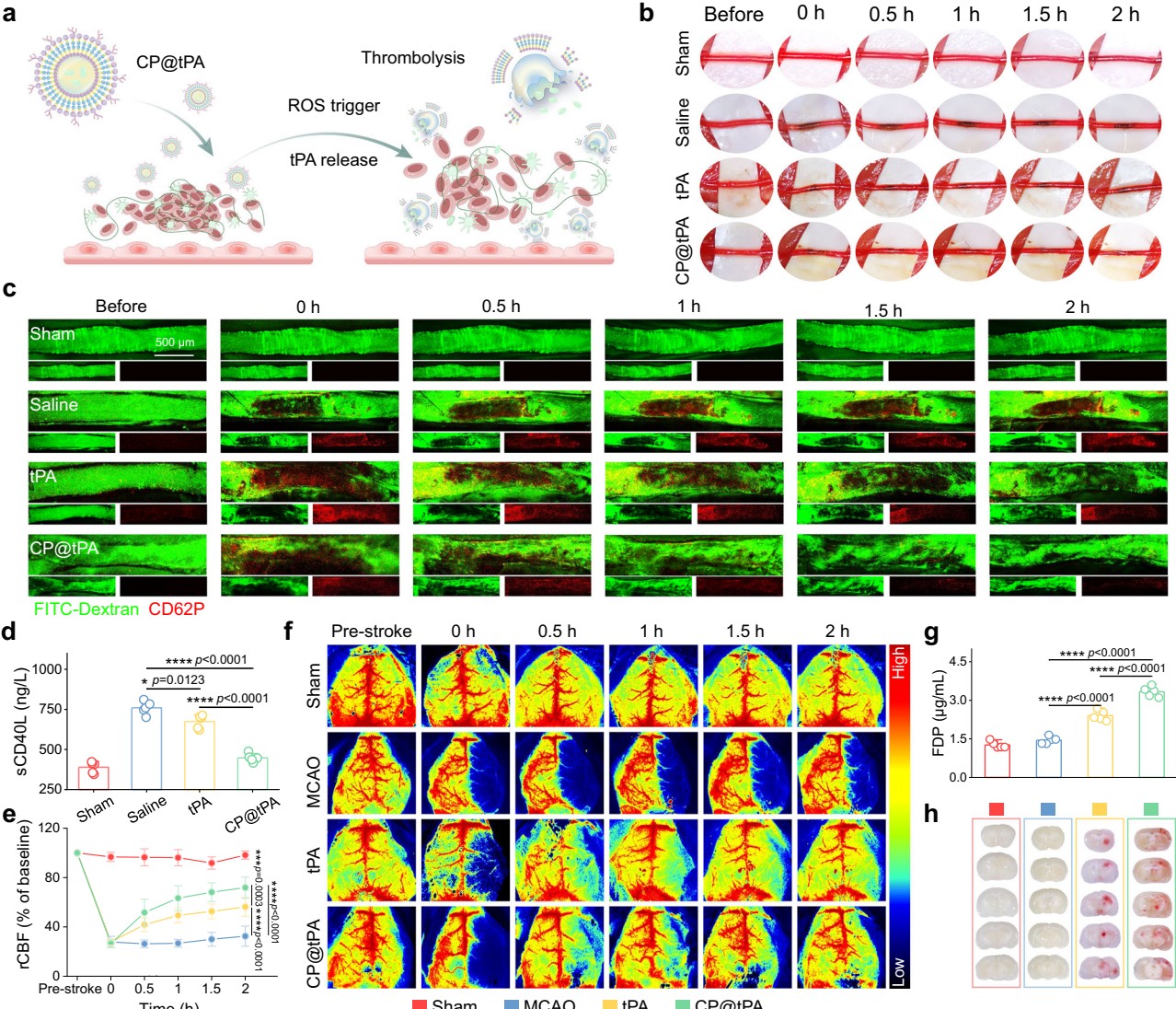

**Fig. 3 | In vivo thrombolytic efficacy of CP@tPA polymersomes. a** Schematic illustration of thrombolysis by CP@tPA polymersomes. **b** Digital photographs showing thrombolysis in lower extremity arterial thrombosis model. Thrombolytic therapy was initiated 10 min after model establishment. **c** In vivo two-photon imaging showing thrombolysis in lower extremity arterial thrombus model. Blood vessels were imaged by intravenous injection of FITC-dextran (MW = 2000 kDa, green). Thrombus were labeled with activated platelets (CD62P+, red). **d** ELISA measurement of plasma sCD40L levels in mice with lower extremity arterial thrombus model 2 h after thrombolysis (*n* = 5 biologically independent samples).

**e** rCBF quantification of embolic MCAO model in different treatment groups (*n* = 5 biologically independent samples). **f** Laser speckle contrast images of MCAO model in different treatment groups. **g** ELISA measurement of plasma FDP levels in mice 2 h after thrombolysis (*n* = 5 biologically independent samples). **h** Digital photos showing cerebral hemorrhage in MCAO mice 22 h after thrombolysis. The results in (**c**) were representative of three independent experiments. Data in (**d**, **e**,**g**) were presented as mean ± SD. *P* values were calculated by one-way ANOVA and two-way followed by Bonferroni's post-hoc test (**d**, **e**, **g**). Source data are provided as a Source Data file.

(Fig. 3d). This dual therapeutic effect (ROS scavenging by CP polymersomes and enhanced tPA delivery) likely underlies the observed thrombolytic enhancement. Laser speckle imaging confirmed superior perfusion recovery in CP@tPA-treated subjects. Lower extremity arterial thrombi showed 78.15 ± 9.44% flow restoration versus 37.15 ± 6.76% (saline) and 57.12 ± 6.39% (tPA) (Supplementary Fig. 14a, b). Hematoxylin and eosin (H&E) staining of vascular sections at the thrombus site revealed thrombus areas in CP@tPA group were reduced to 26.53% of saline group (Supplementary Fig. 14c, d). Comparable efficacy advantages were observed in carotid artery thrombus model (Supplementary Fig. 15a–c).

We further evaluated the thrombolytic efficacy of CP@tPA using the MCAO model, a well-established preclinical model for studying ischemic stroke. Thrombolytic therapy was initiated 2 h poststroke, and regional cerebral blood flow (rCBF) was continuously monitored.

Thrombus injection into the middle cerebral artery led to a precipitous drop in rCBF, with flow persistently restricted to less than 30%, mimicking the severe ischemic conditions observed in stroke patients (Fig. 3e, f). 2 h after administration, free tPA restored partial reperfusion, with rCBF recovering to approximately 50%. In contrast, CP@tPA treatment achieved significantly superior recovery, with rCBF reaching 71.86 ± 8.57%. To quantify thrombus dissolution, we measured serum levels of fibrin degradation products (FDP), a biomarker of fibrinolysis[34]. As shown in Fig. 3g, FDP level in the CP@tPA group surged compared to the MCAO and tPA groups. The superior thrombolytic efficacy of CP@tPA can be attributed to its unique properties, including prolonged blood half-life, thrombus targeting ability, and ROS-responsive drug release characteristics. These features collectively enhance the bioavailability and therapeutic effect of tPA at the site of thrombosis.

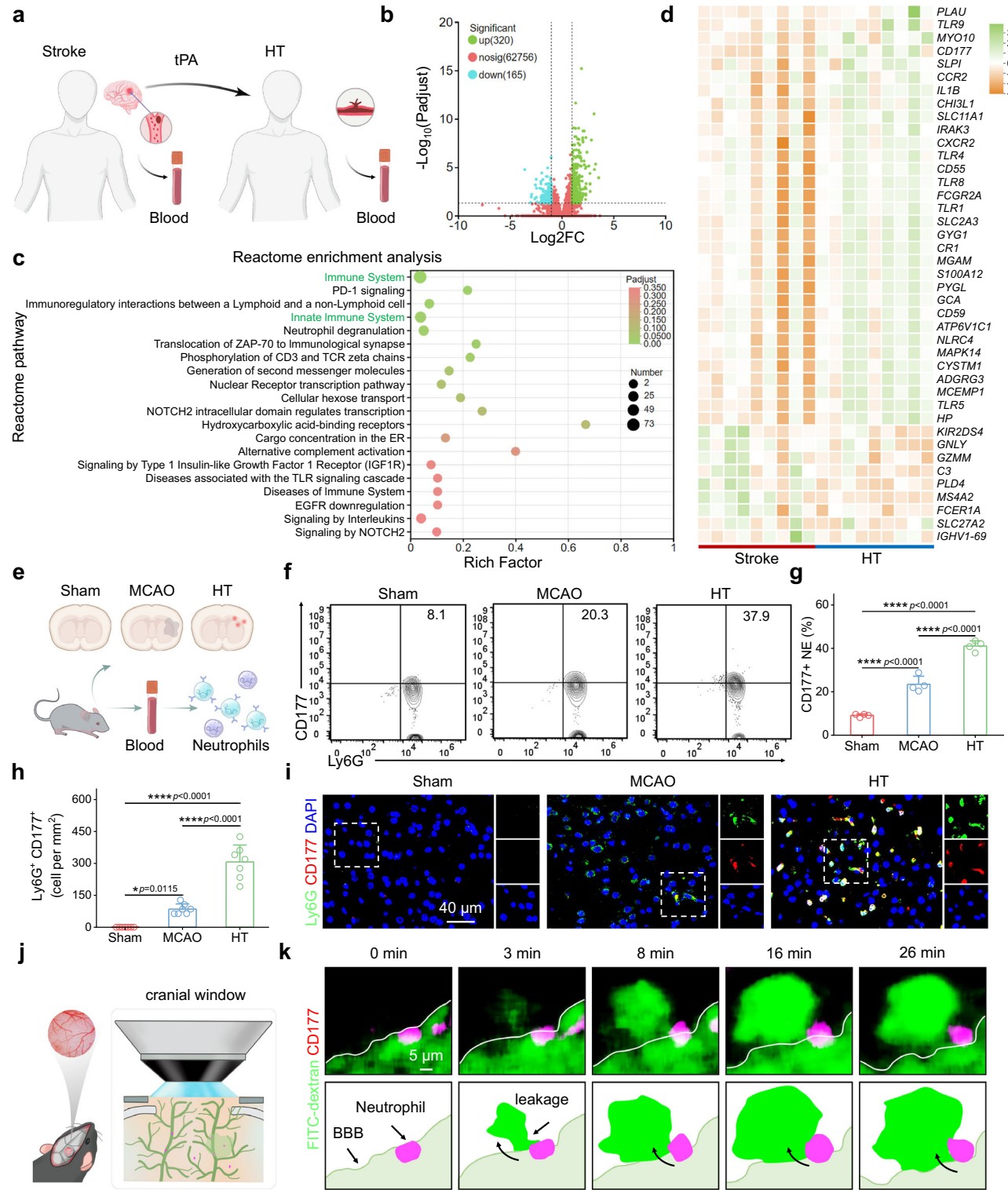

## CD177+ neutrophils induce hemorrhagic transformation

Despite the superior thrombolytic efficacy demonstrated by CP@tPA, it has not effectively alleviated the HT side effect after thrombolysis (Fig. 3h and Supplementary Fig. 16). To elucidate the mechanisms underlying tPA-induced HT, we collected peripheral blood samples from ischemic stroke patients who developed HT following thrombolysis (Fig. 4a). RNA sequencing (RNA-seq) analysis revealed significant changes in the expression of 485 genes (320 upregulated and 165 downregulated) in HT patients (Fig. 4b). Pathway enrichment analyses using Kyoto Encyclopedia of Genes and Genomes (KEGG) and

Reactome indicated that tPA significantly impacts pathways related to the immune system, innate immune system and inflammatory signaling, including MAPK, NF-κB, Toll-like receptors signaling pathways, and Th17 cell differentiation (Fig. 4c and Supplementary Fig. 17). Notably, the KEGG analysis detected an enrichment of genes associated with NET formation, suggesting a potential association between tPA-induced HT and neutrophils. By analyzing the GO enrichment of differentially expressed genes within the innate immune system, these genes are involved in neutrophil activation, particularly through the upregulation of *CXCR2* and *CD177* (Fig. 4d), as well as their

**Fig. 4 | CD177$^+$ neutrophils are associated with tPA-induced HT. a** Schematic illustration of clinical sample acquisition. Peripheral blood was collected from stroke patients before tPA therapy and patients with HT after tPA thrombolysis. **b** Volcanic of transcriptome changes between Stroke and HT groups. Differentially expressed genes (DEGs) were identified using DESeq2 (|log$_2$(fold change, FC)| ≥ 1 and FDR ≤ 0.05). **c** Reactome enrichment analysis of the differentially-expressed genes between Stroke and HT groups. Analyzed using a two-sided Fisher's exact test with Benjamini–Hochberg correction for multiple comparisons. **d** Heatmap showing differential genes related to the innate immune system. **e** Illustration of mouse sample acquisition. MCAO mice were intravenously injected with tPA at 2 h poststroke. 22 h later, peripheral blood and brain tissues were collected from mice that developed HT. **f** Representative flow cytometry plot of CD177$^+$Ly6G$^+$ cells in mouse blood. **g** Percentages of CD177$^+$Ly6G$^+$ cells according to flow cytometry analysis (n = 4 biologically independent samples). **h** The quantitative analysis of CD177-expressed neutrophils (CD177$^+$Ly6G$^+$) from immunofluorescence imaging

(n = 7 biologically independent samples). **i** Representative immunofluorescence staining images showing CD177-expressed neutrophils (red) in mouse brain. Neutrophil were labeled with Ly6G (red), and the nuclei were stained with DAPI (blue). **j** Schematic illustration showing the detection process of CD177$^+$ neutrophils penetrating the BBB through two-photon microscopy. The mouse head was secured in a stereotaxic frame for cranial window and MCAO model establishment. Thrombolytic therapy was administered at 2 h poststroke, and two-photon imaging was performed at 10 h after treatment. **k** Two-photon imaging showing the BBB penetration process of CD177$^+$ neutrophils. The migrated cells caused vascular leakage simultaneously. Green, FITC-dextran (MW = 40 kDa); purple, APC-CD177 labeled CD177$^+$ neutrophils. The results in (**k**) were representative of three independent experiments. Data in (**g**, **h**) were presented as mean ± SD. P values were calculated by one-way ANOVA followed by Bonferroni's post-hoc test (**g**, **h**). Source data are provided as a Source Data file.

participation in various immune response pathways (Supplementary Fig. 18). Notably, *CD177* expression in the HT group was 16.6 times higher than that in the stroke group (Supplementary Fig. 19). Previous studies have proved that CD177 can bind with CD31 to promote neutrophil transmigration[18–21].

To further elucidate the potential relationship between CD177$^+$ neutrophils and tPA-induced HT, we performed flow cytometry and immunofluorescence staining on peripheral blood and brain tissues from tPA-treated MCAO mice that developed HT (Fig. 4e). The number of CD177$^+$ neutrophils in the HT mouse blood was 1.8-fold higher compared to that in the MCAO group (Fig. 4f, g and Supplementary Fig. 20). Moreover, a marked increase in CD177$^+$ neutrophil infiltration was observed in the brains of HT mice, reaching levels approximately 3.6 times higher than those in the MCAO group (Fig. 4h, i). The expression of CD177 can also be upregulated when treating the isolated mouse neutrophils with tPA (Supplementary Fig. 21). These findings, combined with clinical data, suggest that tPA-induced HT involves the upregulation of CD177$^+$ neutrophils and their infiltration into brain tissue. The process of neutrophil migration may contribute to BBB disruption, which could serve as a critical prerequisite for HT development. To further investigate the role of CD177$^+$ neutrophils in BBB permeability, we created a cranial window above the ischemic brain to monitor the real-time dynamics of CD177$^+$ neutrophil transmigration and vascular leakage (Fig. 4j). Following tPA administration, CD177$^+$ neutrophils were initially adhering to the luminal side of the vascular wall, and subsequently infiltrated into the ischemic brain in a time-dependent manner (Fig. 4k). Concurrently, we observed that as CD177$^+$ neutrophils migrated from the vasculature into brain tissue, they induced BBB disruption, leading to fluorescein iso-thiocyanatedextran (FITC-dextran) leakage. The extravasation was significantly enhanced in regions where CD177$^+$ neutrophil infiltration occurred (Supplementary Fig. 22).

The increased BBB permeability mediated by CD177$^+$ neutrophils is likely due to their enhanced ability to generate NETs, which play a pivotal role in tPA-induced HT. Specifically, histones and proteases within NETs directly degrade tight junction proteins and the basal lamina, leading to increased vascular permeability[35,36]. Additionally, NETs enhance thrombo-inflammation by activating platelets and amplifying the release of pro-inflammatory cytokines, further compromising BBB integrity[37]. To prove this, we evaluated NET formation after 90 min of tPA stimulation. Neutrophils triggered by tPA displayed markedly intensified SYTOX signals compared to the MCAO group, confirming enhanced NET generation (Supplementary Fig. 23a). Quantitative analysis using enzyme-linked immunosorbent assay (ELISA) further supported these observations, with NET and MPO levels in the tPA group elevated by 2.2-fold and 1.9-fold, respectively, relative to the MCAO group (Supplementary Fig. 23b, c). These findings collectively uncover a critical pathway underlying HT: tPA drives CD177$^+$ neutrophil recruitment to the brain, where excessive NET

production compromises BBB integrity, culminating in hemorrhagic complications.

## CP@rCD177 polymersomes improved tPA-induced HT

Intervention against CD177$^+$ neutrophil infiltration represents a promising therapeutic strategy to mitigate HT. Given that CD177$^+$ neutrophils interact with CD31 through surface receptors, an interaction blockable by rCD177[38], we evaluated rCD177's efficacy in suppressing neutrophil migration using transwell assays (Fig. 5a). By counting the cell number in the lower chamber, the result revealed that rCD177 reduced neutrophil migration rates by 34.8% (from 78.4% to 51.1%, Fig. 5b). Flow cytometry further demonstrated a 42.5% decrease in tPA-induced CD177$^+$ neutrophil transmigration with rCD177 treatment (Fig. 5c and Supplementary Fig. 24). Notably, NET production in the lower chamber was dramatically reduced to 23.4% of tPA-treated group levels (Supplementary Fig. 25). These findings position rCD177 as a potential therapeutic agent that attenuates CD177$^+$ neutrophil-mediated BBB disruption, thereby ameliorating HT progression.

To enhance therapeutic precision, we engineered CP@rCD177 by encapsulating rCD177 into CP polymersomes. CP@rCD177 possessed the particle size of 124.3 nm and zeta potential of -3.85 ± 0.75 mV (Supplementary Table 3). The drug loading content and encapsulation efficiency of CP@rCD177 are 6.62% and 72.82%, respectively. Similarly, CP@rCD177 also exhibit excellent colloidal stability and long-term particle stability (Supplementary Fig. 26a, b), comparable to that of CP@tPA (Supplementary Fig. 6a, b). This nanoplatform achieves ROS-triggered payload release (Supplementary Fig. 26c, d) and prolonged systemic circulation (3.88-fold half-life extension vs free rCD177; Supplementary Fig. 26e and Supplementary Table 4), ensuring sustained bioactivity in vivo. We next assessed CP@rCD177's capacity to inhibit brain infiltration of CD177$^+$ neutrophils (Fig. 5d). Immunofluorescence at 22 h post-tPA administration revealed perivascular CD177$^+$ neutrophil accumulation in mouse brains (Fig. 5e). When mice were pre-treated with CP@rCD177 before thrombolysis, there was a marked overlap between CD31 and rCD177 at the vascular interfaces, along with a substantial reduction in the number of CD177$^+$ neutrophils in the brain tissue (Fig. 5f). Flow cytometry analysis demonstrated that either tPA or CP@tPA treatment upregulated the neutrophil populations (Fig. 5g). In contrast, interventions with rCD177 and CP@rCD177 significantly reduced this occurrence. Notably, CP@rCD177 therapy decreased cerebral neutrophils by 82.5% and CD177$^+$ neutrophils by 87.4% compared to the tPA group (Supplementary Fig. 27a–c). Complementary immunofluorescence further validated CD177$^+$ neutrophil suppression with CP@rCD177 intervention (Supplementary Fig. 28). These findings establish that CP@rCD177-mediated CD31 blockade disrupts neutrophil-endothelial interactions, effectively preventing CD177$^+$ neutrophil transmigration into ischemic brain.

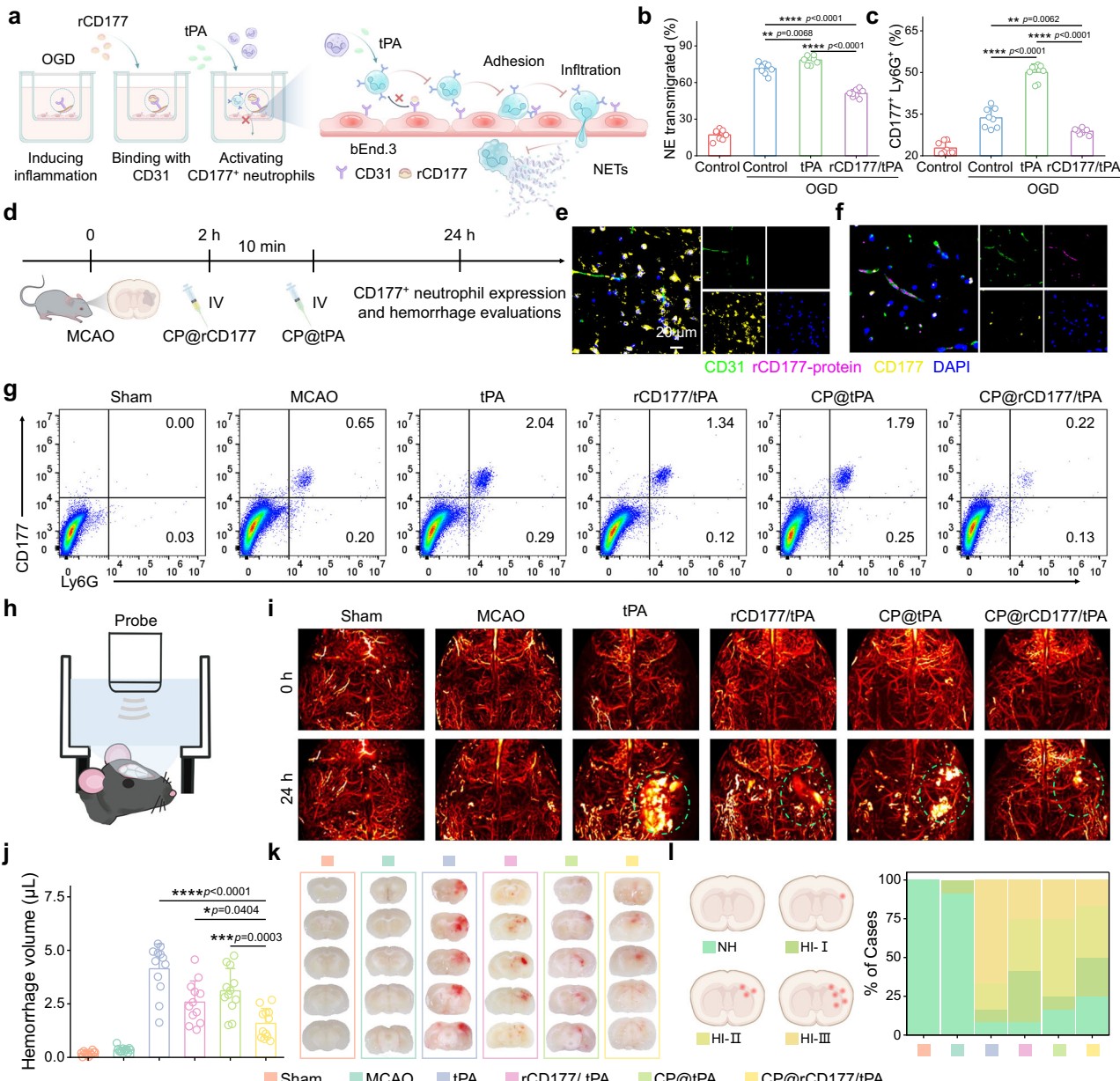

**Fig. 5 | CP@rCD177 polymersomes improved tPA-induced HT. a** Schematic illustration of rCD177 inhibiting CD177⁺ neutrophils migration using in vitro BBB model. bEnd3 cells were inoculated into the upper chamber of transwell to establish in vitro BBB model. After 3 h of OGD processing, rCD177 was added for 10 min of incubation. Then neutrophils isolated from MCAO mouse blood along with tPA were added, 90 min later, cells in the lower chamber were counted and analyzed by flow cytometry. **b** Mobility of neutrophils in the lower chamber according to transmigration assay (n = 8 biologically independent samples). **c** Percentage of CD177⁺Ly6G⁺ cells in the lower chamber (n = 8 biologically independent samples). **d** Schematic illustration of CP@rCD177 intervention before thrombolysis. CP@rCD177 was administered at 2 h after MCAO, followed by CP@tPA injection 10 min later. At 24 h poststroke, CD177⁺ neutrophil expression and hemorrhagic transformation were investigated. Representative immunofluorescence staining images showing the reduced accumulation of CD177⁺ cells (yellow) in the infarct brain 22 h after tPA (**e**) and CP@rCD177/tPA (**f**) treatment. rCD177 was labeled with FITC (purple), vessels were labeled with CD31 (green), and nuclei were stained with DAPI (blue). **g** Representative flow cytometry plot of CD177⁺Ly6G⁺ neutrophils in mouse brains 22 h after treatment. **h** A schematic diagram of observing mouse brain hemorrhage using photoacoustic microscopy. **i** Representative photoacoustic imaging showing reduced brain hemorrhage 22 h after CP@rCD177/tPA treatment. **j** Cerebral hemorrhage volume in different groups (n = 12 biologically independent samples). Green circles mark the bleeding sites. **k** Digital photos showing cerebral hemorrhage in MCAO mice. **l** Cerebral hemorrhage classification in different groups (n = 12 biologically independent samples). NH: no hemorrhage, HI-I: single petechia, HI-II: 2 - 3 petechiae, HI-III: more than 3 petechiae. The results in (**e**, **f**) were representative of three independent experiments. Data in (**b**, **c**, **j**) were presented as mean ± SD. P values were calculated by one-way ANOVA followed by Tukey's post-hoc test. Source data are provided as a Source Data file.

To further investigate the potential of CP@rCD177 in preventing hemorrhagic complications, we evaluated cerebral hemorrhage 22 h after therapy. Photoacoustic imaging (PA) was first employed for real-time monitoring of intracerebral bleeding (Fig. 5h). Capitalizing on the strong optical absorption characteristic of hemoglobin at 532 nm[39], we utilized PA signals at this specific wavelength for reconstruction of cerebral vasculature. PA observation revealed distinct spatial distribution patterns of hemoglobin signals across groups (Fig. 5i). The sham and MCAO groups exhibited signals predominantly confined within the vascular architecture, whereas thrombolytic treatment

groups (tPA and CP@tPA) demonstrated significant extravasation of hemoglobin signals into the parenchymal tissue, indicative of HT. Notably, therapeutic intervention with rCD177 formulations substantially attenuated hemoglobin signal dispersion, with CP@rCD177 therapy showing the most pronounced protective effect against HT. Quantitative assessment of hemorrhagic volume using a hemoglobin detection kit demonstrated that CP@rCD177 pretreatment significantly reduced bleeding by 61.6% and 48.7% compared to tPA and CP@tPA groups, respectively (Fig. 5j). Examination of brain sections corroborated these findings, revealing differential HT severity across treatment groups (Fig. 5k). We then investigated the hemorrhage severity based on petechiae quantification in the ischemic hemisphere[40]. While the overall hemorrhage rates were 91.7% and 83.3% in tPA and CP@tPA groups, respectively, CP@rCD177 pretreatment reduced this incidence to 75% (Fig. 5l). More specifically, compared with the CP@tPA group, the intervention demonstrated particular efficacy in mitigating severe hemorrhage, with HI-II and HI-III classifications both showing a 33.3% reduction. Concurrently, we observed a 50% increase in NH (no hemorrhage) cases and a 200% elevation in HI-I (mild hemorrhage) instances.

Neutrophil infiltration is closely associated with endothelial dysfunction and increased BBB permeability; blocking this infiltration has been shown to effectively attenuate post-stroke BBB injury[5]. To determine whether CP@rCD177 could confer similar protective effects, we investigated its capacity to ameliorate neutrophil-mediated BBB damage. First, Evans Blue (EB) extravasation assay was performed. The brain sections showed that cerebral ischemia induced substantial EB leakage (Supplementary Fig. 29a), confirming BBB structural compromise. Notably, MCAO mice subjected to thrombolytic therapy exhibited a marked elevation in BBB permeability, with spatial colocalization between dye extravasation zones and hemorrhagic regions. While free rCD177 administration showed moderate attenuation of BBB disruption, residual EB leakage remained significant compared to sham group, a phenomenon potentially attributable to the short blood half-life (Supplementary Fig. 26e and Supplementary Table 4). Notably, pretreatment with CP@rCD177 nanoparticles before CP@tPA thrombolysis achieved superior BBB protection, reducing EB extravasation to 66.9% of thrombolysis-only control (Supplementary Fig. 29b). To further assess BBB integrity in the ischemic hemisphere, we employed intravital microscopy and TEM. Intravital imaging confirmed pronounced vascular leakage in both thrombolysis-treated and MCAO groups relative to sham controls (Supplementary Fig. 30). While pretreatment with CP@rCD177 significantly reduced this extravasation. TEM analysis further demonstrated disruption of the BBB ultrastructure following HT and ischemia, characterized by loss of tight junctions and formation of inter-endothelial gaps (Supplementary Fig. 31). Importantly, administration of CP@rCD177 substantially reversed these pathological changes, reducing the number of inter-endothelial gaps to approximately 49% of those in the CP@tPA group. This protective effect may play a critical role in suppressing hemorrhagic complications and maintaining neurovascular function.

Triphenyltetrazolium chloride (TTC) staining was then performed to evaluate the effect of nanodrugs on ischemic damage. MCAO mice developed substantial cerebral infarction, exhibiting an infarct volume of 35.68% at 24 h poststroke (Supplementary Fig. 32). Administration of free tPA and CP@tPA reduced the infarct volume to 28.07% and 19.91%, respectively. Notably, intervention with CP@rCD177 resulted in a further 39.21% reduction in infarct volume compared to CP@tPA group, culminating in a final infarct volume of only 12.1%. This outcome underscores its superior efficacy in mitigating ischemic brain damage by inhibiting CD177+ neutrophils infiltration. The therapeutic efficacy of CP@rCD177/tPA combination therapy was further substantiated by a significant improvement in survival outcomes in MCAO mice. Through its ability to stabilize the

BBB and mitigate brain infarct, CP@rCD177 intervention enhanced the 24-h survival rate to 66.67% (Supplementary Fig. 33). These comprehensive findings establish that CP@rCD177 pretreatment serves as an effective neuroprotective strategy, simultaneously reducing the HT incidence and brain damage while significantly attenuating thrombolysis-associated mortality risk.

## CP@rCD177 polymersomes alleviated neuroinflammation and improved long-term neurological effects

To assess post-treatment neuroinflammation, key pro-inflammatory cytokines (IL-1β, TNF-α) and anti-inflammatory cytokines (IL-10, TGF-β) in the ischemic hemispheres were measured on day 3 after treatment. As shown in Supplementary Fig. 34, both ischemia (MCAO group) and HT (tPA and CP@tPA groups) resulted in significantly elevated pro-inflammatory cytokines and reduced anti-inflammatory mediators. This inflammatory response is likely triggered by ischemia-reperfusion injury, tPA-induced neutrophil chemotaxis, and microglial activation[41–44]. Importantly, CP@rCD177 therapy significantly attenuated this response, reducing IL-1β and TNF-α levels by 12.1% and 48.5%, respectively, while increasing IL-10 and TGF-β by 28.6% and 39.6%, respectively, compared to the MCAO group. These results suggest that targeting CD177+ neutrophils can effectively alleviate thrombolysis-related neuroinflammation.

We next assessed the long-term neurobehavioral recovery in mice through a comprehensive behavioral test, which incorporated multiple sensorimotor assessments including the modified neurological severity score (mNSS), corner turn, rotarod, and adhesive removal tests, as well as the Barnes maze test to evaluate cognitive function. Our results revealed that both MCAO surgery and post-thrombolysis led to severe motor deficits, accompanied by significantly elevated mNSS scores (Supplementary Fig. 35). The functional impairments associated with tPA thrombolysis were closely correlated with the hemorrhagic complications it induced[14,45,46]. In contrast, pretreatment with CP@rCD177 prior to thrombolysis markedly improved functional outcomes: reduced mNSS scores, more symmetrical turning behavior, shorter removal time in the adhesive test, and longer latency to fall on the rotarod (Supplementary Fig. 36a–g). Spatial memory was assessed using the Barnes maze on day 28 post-treatment (Supplementary Fig. 36h). Mice in the MCAO, tPA, and CP@tPA groups required significantly more time, traveled longer distances, and made more errors before locating the escape box compared to sham mice (Supplementary Fig. 36i–k). CP@rCD177 treatment effectively attenuated these deficits, indicating preserved cognitive function. Together, these results demonstrate that the combine use of CP@rCD177 and CP@tPA promotes significant recovery of sensorimotor and cognitive functions, highlighting its potential as a promising therapeutic strategy for ischemic stroke.

## CP@rCD177 inhibited HT by reducing CD177+ neutrophils-associated NET formation

To elucidate the mechanistic basis of CP@rCD177-mediated HT inhibition, we performed transcriptome profiling of infarct hemispheres 24 h post-MCAO. Ischemia induced substantial transcriptomic changes with 830 DEGs (221 upregulated, 609 downregulated) compared with sham control (Supplementary Fig. 37). Only slight alterations were detected between tPA-mediated HT group and MCAO group, with 65 DEGs identified (14 upregulated and 51 downregulated), suggesting shared molecular injury mechanisms between ischemic and hemorrhagic pathologies. The combination therapy of CP@rCD177/tPA elicited a distinct genomic signature, reversing the HT-induced expression pattern with 138 DEGs (127 upregulated, 11 downregulated). Heatmap revealed that CP@rCD177/tPA significantly suppressed neutrophil activation pathways, including chemotaxis, migration, while also markedly downregulating NET formation genes (Fig. 6a, b). KEGG enrichment analysis further demonstrated the multimodal action of

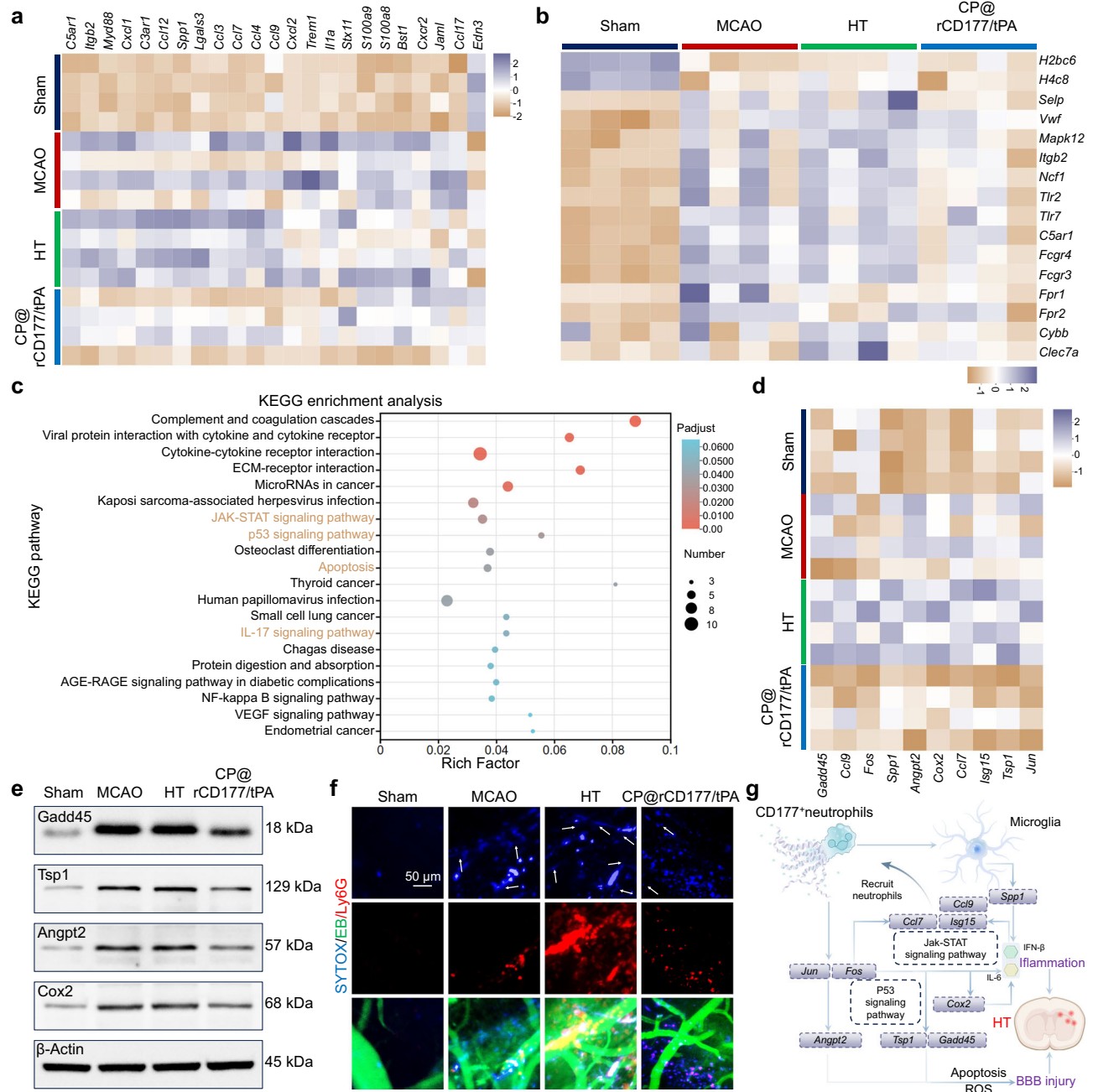

**Fig. 6 | The mechanistic insights of CP@rCD177-mediated HT inhibition.**
**a** Heatmap showing differential genes associated with neutrophil chemotaxis, activation and infiltration. **b** Heatmap showing differential genes associated with NET formation. **c** KEGG enrichment analysis of the differential genetic pathways between HT and CP@rCD177/tPA groups. Analyzed using a two-sided Fisher's exact test with Benjamini–Hochberg correction for multiple comparisons. **d** Changes in, *Gadd45, Ccl9, Fos, Spp1, Ccl7, Isg15, Jun, Angpt2, Cox2,* and *Tsp1* gene expression among different groups. **e** Western blot analysis of GADD45, TSP1, ANGPT2 and COX2 in different groups 24 h poststroke. **f** Representative immunofluorescence staining images showing neutrophil infiltration (Ly6G⁺, red), NET distribution (SYTOX⁺, blue) and vascular leakage (EB, green) in mouse brains 22 h after therapy. White arrows indicate NETs. **g** Schematic diagram of CP@rCD177 inhibiting hemorrhagic complications associated with CD177⁺ neutrophils-NETs. The results in (**e, f**) were representative of three independent experiments. Source data are provided as a Source Data file.

CP@rCD177/tPA through modulating apoptosis regulation (p53), inflammatory cascades (JAK-STAT, IL-17) (Fig. 6c). Subsequently, DEGs associated with BBB disruption were selected for heatmap analysis (Fig. 6d). The results revealed a marked upregulation of key inflammatory and apoptotic regulators *Jun/Fos* in the HT group[47,48], and genes linked to apoptosis (*Gadd45*), oxidative stress (*Tsp1*), and inflammation (*Cox2*) were elevated as well[49–51]. Notably, chemokines (*CCL7, CCL9*) and the interferon-stimulated gene *Isg15* were also upregulated, suggesting enhanced neutrophil recruitment to the injury site[52–54]. Furthermore, the increased expression of *Angpt2* likely contributes to heightened vascular permeability[55]. These transcriptional changes reflect dynamic BBB damage-repair mechanisms that critically influence neuroinflammatory states and vascular stability. CP@rCD177 intervention counteracted these effects, suppressing the expression of *AP-1* and its downstream targets, with concordant protein-level validation via Western blot (Fig. 6e). Furthermore,

CP@rCD177 attenuated HT-induced inflammatory cytokines IFN-β and IL-6, underscoring its anti-inflammatory efficacy (Supplementary Fig. 38).

Integrating these findings with NET dynamics, we identified a mechanistic link between HT and NET formation. By intravital microscopy monitoring, sham-operated mice exhibited intact vasculature with minimal NETs, while excessive NETs were generated in HT group— predominantly localized at sites of vascular leakage and surrounded by abundant neutrophils (Fig. 6f). This observation suggests that neutrophil-derived NETs actively contribute to BBB destabilization. Strikingly, CP@rCD177/tPA reduced cerebral NET content by 51.8% compared to tPA monotherapy (Supplementary Fig. 39), likely due to pre-thrombolytic CP@rCD177 administration limiting CD177+ neutrophil recruitment. Immunofluorescence mapping revealed NETs in tPA-induced HT brains predominantly co-localized with microglia (Iba1+), with minimal association with astrocytes (GFAP+) or neurons (NeuN+) (Supplementary Fig. 40). This spatial specificity implies CD177+ neutrophil-derived NETs exacerbate neuroinflammation via microglial activation[44]. Collectively, tPA-induced HT involves a sequential pathological cascade initiated by the infiltration of CD177+ neutrophils, which drives excessive NET production, subsequently triggering microglia-mediated inflammation, and ultimately leading to BBB disruption and blood effusion (Fig. 6g). Preemptive CP@rCD177 administration disrupts this pathogenic axis, preserving BBB integrity and mitigating HT.

## Discussion

The development of thrombolytic therapies that mitigate hemorrhagic complications remains a critical unmet need in ischemic stroke management[56]. Our study introduces a dual nanotherapeutic strategy combining fibrin-targeted, ROS-responsive polymersomes (CP@tPA) with CD177+ neutrophil interception (CP@rCD177), effectively addressing both clot resolution and HT prevention. This approach leverages the synergistic benefits of targeted drug delivery and immunomodulation, offering a valuable therapeutic strategy in tPA-based stroke therapy.

A key limitation of tPA is its propensity to exacerbate BBB disruption and neutrophil-mediated inflammation, leading to HT. We previously have implicated NETs as central drivers of thromboinflammation and BBB damage post-reperfusion[57]. Our current findings extend this understanding by identifying CD177+ neutrophil-derived NETs as critical mediators of BBB disruption, with spatial colocalization of NETs and microglial activation zones. By competitively blocking CD31 with CP@rCD177, we disrupted neutrophil-endothelial adhesion, reducing cerebral infiltration by 87.4% and NET production by 51.8%. This aligns with recent works, which showed that CD177 inhibition attenuates neutrophil migration in acute lung injury and acute pancreatitis models[58,59], but our study uniquely applies this strategy to cerebrovascular pathology.

The design of CP polymersomes builds upon advances in stimuli-responsive nanomedicine. ROS-triggered systems have been described for ischemic stroke therapy[60–62], highlight the potential of redox-sensitive drug delivery. Our fibrin-targeted platform enhances site-specific drug release, achieving 5.14-fold greater thrombus accumulation than non-targeted counterparts. Compared with the earlier thrombus-homing systems, such as fibrinogen-mimicking nanovesicle developed by Huang et al.[63] CP polymersomes exhibit superior properties by integrating CREKA peptide with ROS sensitivity for spatiotemporal control. Furthermore, CP polymersomes extended the half-life of tPA by 9.4 times, addressing a longstanding pharmacokinetic limitation highlighted in clinical studies[64]. While CP@tPA administration substantially enhanced thrombolytic efficacy, it also resulted in more extensive vascular leakage and did not mitigate hemorrhagic complications. In contrast, pretreatment with CP@rCD177 markedly attenuated this pathological extravasation. Ultrastructural analysis

further confirmed that CP@rCD177 effectively restored BBB integrity, reducing the number of inter-endothelial gaps to approximately 49% of that observed in the CP@tPA group.

The transcriptional profiling of HT brains revealed AP-1-driven upregulation of BBB-disrupting genes (e.g., *Angpt2, Cox2*), consistent with reports linking AP-1 to neuroinflammation[65–67]. CP@rCD177 reversed this signature through specific neutrophil-mediated pathways. Notably, our combination therapy uniquely suppressed NETs-associated microglial activation, a phenomenon implicated in tPA-induced brain hemorrhage[43]. This thrombolysis and immunomodulation dual action mirrors emerging trends in combinatorial stroke therapies.

In conclusion, our work establishes CD177+ neutrophil interception as a viable strategy to expand tPA's therapeutic window. By integrating fibrin-targeted thrombolysis with immunomodulatory nanotherapy, we hold promise in addressing decades-old clinical challenge, offering a potential method for precision medicine in cerebrovascular diseases.

## Methods

### Materials

Trizol was purchased from TakaRa (Japan). D,L-Lactide (D,L-LA), 4-Carboxyphenylboronic acid (PBA), 1,1,1-Tris(hydroxymethyl)ethane (TME), anhydrous toluene and tetrahydrofuran (THF) were purchased from Adamas (China). Dichloromethane (DCM) was purchased from Heowns (China). Sn(Oct)$_2$, 4-Dimethylaminopyridine (DMAP), Dicyclohexylcarbodiimide (DCC), Evans blue, ICG, and FITC-dextran were purchased from Sigma-Aldrich (China). Mal-PEG-OH was purchased from Ponsure (China). CREKA peptide was purchased from the Chinese Peptide Company (China). Calcein acetoxymethyl ester/Propidium iodide (Ca-AM/PI), CCK-8, and Hemoglobin test kits were purchased from Beyotime Biotechnology (China). Mouse peripheral blood neutrophil isolating fluid kits were purchased from Servicebio (China). TTC and Mouse TNF-α, TGF-β ELISA kit were purchased from Solarbio (China). tPA were purchased from Boehringer Ingelheim (Germany). Hydrogen peroxide assay kit, BCA protein assay kit were purchased from Bioss (China). ELISA kits including sCD40L, ROS, and MPO were purchased from Saipei (China). ELISA kits including IL-6, IFN-β, and FDP were purchased from Shanghai Enzyme-linked (China). Mouse NET and IL-1β ELISA kits were purchased from Jiangsu Meimian (China). Mouse IL-10 ELISA kit was purchased from Multi Sciences (Lianke) Biotech (China). SYTOX Green was purchased from Invitrogen (USA).

### Cells and animals

bEnd.3 cell lines (iCell Bioscience Inc, Shanghai, China) were separately cultivated in DMEM medium (Hyclone) at 37 °C in a humidified atmosphere with 5% CO$_2$. Neutrophils were extracted from the blood of male C57BL/6J mice.

C57BL/6 J mice (8–12 weeks, male) were purchased from Beijing Huafukang Biotechnology Co., Ltd. (Beijing, China). All animals were maintained under 12 h of light-dark cycle at 25 ± 1 °C and 50–60% of humidity with access to food and water ad libitum. All animal tests were approved by the Institutional Animal Care and Use Committee of Southwest Jiaotong University (SWJTU-2203-NSFC(007)).

### Mouse embolic MCAO model[68]

The blood clot was first prepared in vitro. Mouse blood was drawn into a PE-50 tube, coagulated at room temperature for 2 h, and then left at 4 °C for 22 h. The tube containing thrombus was then cut into 5 cm in length, and one side was connected to PE-10 tube attached with another PE50 tube. Saline was repeatedly injected into the syringe for rinse. Then 5% isoflurane was applied to C57BL/6 J mice by gas anesthesia. After deep anesthesia, the carotid artery (CCA), external carotid artery (ECA), and internal carotid artery (ICA) were isolated. The

modified PE50 (terminal outer diameter <0.2 mm) tube was inserted into the ICA through the ECA. Thrombosis (1.5 cm) was injected with saline to form a central cerebral artery embolism. The cerebral blood flow (CBF) was monitored by RFLSI III Laser Speckle Blood Flow Imaging System (RFLSI III, RWD).

## Mouse carotid artery and lower extremity arterial thrombus models[69]

The carotid artery thrombus and lower extremity arterial thrombus were induced by ferric chloride ($FeCl_3$). For carotid artery thrombus model, mice were anesthetized to expose the CCA blood vessels. Then the filter paper ($4 \times 1$ mm) soaked with 10% $FeCl_3$ solution was applied to the CCA surface and removed 3 min later. After 10 min, thrombosis was observed under the digital microscope (DOM-1001, RWD). For lower extremity arterial thrombus model, mice were anesthetized to expose the femoral artery. Then 10% $FeCl_3$-soaked filter paper ($1 \times 1$ mm) was applied to the vessel surface for 1 min before removal.

## Characterization of thrombus from patients and mice

The thrombus of patients with ischemic stroke and patients with carotid artery embolism was removed by mechanical thrombectomy, and fixed with electron microscopic fixative (2.5% glutaraldehyde). After dehydration, drying, and gold spraying, the thrombus was observed by SEM(JSM-IT700HR, Japan). A similar method was used to observe the thrombus structure in mouse thrombus models.

## Synthesis and characterization of CREKA-PEPLA

Synthesis of Mal-PEG-PBA. Mal-PEG-OH (2 g, 1 mmol), PBA (0.17 g, 1 mmol), DMAP (0.37 g, 3 mmol) and DCC (0.62 g, 3 mmol) were dissolved in a mixture of DCM and THF (v/v, 50:50) under nitrogen atmosphere for 24 h of reaction in dark. DCM and THF were then removed by rotary evaporation. The concentrate was precipitated through petroleum ether and dried under vacuum to obtain Mal-PEG-PBA.

Synthesis of Mal-PEG-PBE. Mal-PEG-PBA (0.22 g, 0.1 mmol) and TME (0.06 g, 0.5 mmol) were dissolved in anhydrous toluene, and heated at 120 °C reflux for 10 h in the Dean-Stark dewatering unit. The mixture was concentrated by rotary evaporation, precipitated with ice ether, and dried under vacuum to obtain Mal-PEG-PBE.

Synthesis of Mal-PEPLA and Mal-PELA. Mal-PEPLA was synthesized through ring-opening polymerization (ROP) of D,L-LA initiated by Mal-PEG-PBE catalyzing using $Sn(Oct)_2$. Briefly, Mal-PEG-PBE (1 g, 0.44 mmol), D,L-LA (4.5 g, 31.25 mmol) and $Sn(Oct)_2$ (0.1 wt%) were quickly added to round-bottom flask with stopcock. Then the flask was degassed under vacuum for 6 h with continuous stirring. The ring-opening polymerization reaction was carried out at 150 °C for 4 h. After the reaction finished, the product was dissolved with a small amount of dichloromethane and precipitated by excess cold ethanol. Mal-PEPLA was obtained by centrifugation and vacuum drying. Mal-PELA was synthesized as above except using Mal-PEG-OH as initiator.

Synthesis of CREKA-PEPLA (CPEPLA). CREKA-peptide (0.05 mmol, 30.2 mg) and Mal-PEPLA (0.05 mmol, 600 mg) were dissolved in dichloromethane at 25 °C under nitrogen atmosphere. After 12 h of reaction, the mixture was precipitated with cold ethanol and dried by lyophilization. Then the freeze-dried powder was dissolved in deionized water and dialyzed (MWCO: 3500) for 24 h, and lyophilized to obtain CPEPLA.

[1]H NMR. The chemical structures of polymers were characterized by [1]H NMR (AM 300, Bruker, Germany). $CDCl_3$ was used as solvent, and Tetra-methylsilane (TMS) was used as the internal reference. Chemical shifts are expressed in parts permillion, ppm.

## Preparation of polymersomes

Polymersomes were fabricated by solvent evaporation method. Briefly, 10 mg of CPEPLA was dissolved in THF (5 mL), and added dropwise into deionized water (5 mL) under stirring. The CPEPLA polymersomes were obtained with the evaporation of THF. Similarly, PELA and PEPLA polymersomes were prepared as above. tPA-loaded polymersomes were prepared by electroporation. 400 μg of polymersomes, 40 μg of rCD177 or tPA, and 360 μL of PBS buffer were added to the electroporation dish (Biorad, USA). Then electroporation was performed at 250 V, 350 μF, and 50 Ω (Biorad, Gene Pulser Xcell, USA). The resulting solution was incubated at 37 °C for 30 min to recover the structure of polymersomes, and centrifuged ($20,000 \times g$, 30 min) to get tPA-loaded polymersomes.

## Characterization of polymersomes

The particle size, polydispersity index (PDI), and zeta potential of the polymersomes were determined by DLS (Zeta-Sizer, Malvern Nano-ZS90, Malvern, Ltd., UK). The vesicle morphology was examined by TEM (TECNAI G2F20FEI, USA). The content of unencapsulated tPA after electroporation was detected by BCA protein assay kit. The drug loading content (DL%) and encapsulation efficiency (EE%) were calculated by the following formula:

$$DL\% = \frac{\text{Weight of drug added} - \text{Unencapsulated drug content}}{\text{Weight of polymersomes}} \times 100\%$$

(1)

$$EE\% = \frac{\text{Weight of drug added} - \text{Unencapsulated drug content}}{\text{Weight of drug added}} \times 100\%$$

(2)

## Colloidal stability and long-term stability[3,70]

To assess the colloidal stability of the polymersomes, CP@tPA and CP@rCD177 (2 mg mL$^{-1}$) were incubated in PBS at 37 °C for 8 h. The size distribution of the polymersomes was measured by DLS at 0, 2, 4, 6 and 8 h. To evaluate the long-term stability of the polymersomes, CP@tPA and CP@rCD177 (2 mg mL$^{-1}$) were stored in PBS at 4 °C in the dark. The size change was determined by DLS at day 0, 7, 15 and 30.

## ROS responsiveness detection

To verify the ROS responsiveness, the polymersomes (100 μg mL$^{-1}$) were treated with $H_2O_2$ (100 μM) for 10 min. The morphology and particle size change of polymersomes was detected by TEM and DLS, and the ROS elimination capacity of polymersomes was determined by hydrogen peroxide kit.

## In vitro drug release

To detect the in vitro release kinetics, tPA-loaded polymersomes were transferred to a dialysis bag (MWCO: 300 kDa), which was then dipped into PBS containing $H_2O_2$ (100 μM) or not, and shake well in a 37 °C bath. The cumulative release of tPA was then detected at the scheduled point of time (0, 0.5, 1, 2, 3, 6, 12, and 24 h) by BCA protein test kit.

## Pharmacokinetics

FITC-labeled tPA and rCD177 were separately encapsulated into CP polymersomes and intravenously injected into mice (doses equivalent to 1 mg kg$^{-1}$ tPA or 1 mg kg$^{-1}$ rCD177). Blood samples were collected at 1, 5, 10, 20, 30, 60, 90, and 120 min post-injection. Plasma was isolated by centrifugation, and the fluorescence intensity of the plasma at each time point was measured using a fluorescence spectrophotometer (Hitachi F-7000, Japan). Drug concentrations in the blood were quantified based on the fluorescence measurements, and pharmacokinetic parameters were calculated using DAS 2.0 software.

## In vitro cytotoxicity study[71]

Cytotoxicity was first evaluated using CCK-8 assay and live/dead staining. For CCK-8 assay, bEnd.3 cells were seeded in 48-well plates at

a density of $5 \times 10^4$ cells per well and allowed to adhere. The cells were then treated with various concentrations of CP@tPA (0, 25, 50, 100, and 200 µg mL⁻¹) for 24 h. After treatment, the medium was replaced with 200 µL of CCK-8 working solution (10% CCK-8 reagent in fresh medium, v/v). The plates were incubated for 0.5 h at 37 °C, after which 100 µL of the resulting solution from each well was transferred to a 96-well plate for absorbance measurement at 450 nm using a microplate reader. Cell viability was calculated by the following formula:

$$\text{Cell vivability}(\%) = \frac{\text{OD}_{450}(\text{Sample}) - \text{OD}_{450}(\text{Blank})}{\text{OD}_{450}(\text{Control}) - \text{OD}_{450}(\text{Blank})} \times 100\% \quad (3)$$

For live/dead staining, bEnd.3 cells were seeded in 24-well plates at a density of $1 \times 10^5$ cells per well and allowed to adhere. The cells were then treated with CP@tPA polymersomes at the specified concentrations. Following 24 h of incubation, the cells were washed with PBS, stained with Ca-AM (2 µM) for 10 min to label live cells, and PI (4 µM) for 10 min to label dead cells. After staining, the cells were washed again to remove excess dye. The stained cells were imaged using a fluorescence microscope (Axio Observer Z1, Zeiss, Germany). In the acquired images, viable cells exhibited green fluorescence, whereas dead cells exhibited red fluorescence.

## Blood compatibility study
Hemolysis test. The hemocompatibility of CP@tPA was first evaluated via hemolysis assay using mouse red blood cells (RBCs). Purified RBCs were diluted with saline to a 2% (v/v) concentration. Then, 0.5 mL of the RBC suspension was mixed with 0.5 mL of CP@tPA at various concentrations (0, 25, 50, 100, and 200 µg mL⁻¹) and incubated at 37 °C for 1 h. Saline and 0.1% Triton X-100 were used as the negative (0% hemolysis) and positive (100% hemolysis) controls, respectively. After incubation, the samples were centrifuged at 300 × g for 10 min. The absorbance of the supernatant was measured at 540 nm using a microplate reader (Elx800, BioTek, USA). The hemolysis rate was calculated using the following formula:

$$\text{Hemolysis ratio}(\%) = \frac{A_s - A_n}{A_p - A_n} \times 100\% \quad (4)$$

$A_s$, $A_n$ and $A_p$ represent the absorption at 540 nm of samples, negative control (saline) and positive control (Triton X-100), respectively.

Coagulation test. 50 µL of recalcified whole blood was introduced into CP@tPA polymersomes with different concentrations (0, 25, 50, 100, and 200 µg mL⁻¹) to initiate coagulation. Deionized water and saline served as the positive and negative controls, respectively. After 1 min, 10 mL of deionized water was added to each sample to lyse any unclotted erythrocytes. The mixtures were then collected and centrifuged at 300 × g for 10 min to isolate the supernatant, which contained hemoglobin released from lysed red blood cells. The absorbance of the supernatant was measured at 540 nm. The BCI was calculated using the following formula:

$$\text{BCI}(\%) = \frac{A_s}{A_p} \times 100\% \quad (5)$$

$A_s$ is the absorbance of the sample and $A_p$ is the positive control absorbance.

## In vitro thrombosis targeting
100 µL of mouse blood was incubated with thrombin (50 U) at 37 °C for 30 min to obtain thrombus. The resulted thrombus was then treated with ICG-loaded PELA (Null@ICG), PEPLA (P@ICG), CPEPLA (CP@ICG) polymersomes, and observed at 15 min, 30 min, 60 min, 90 min, and 120 min using in vivo imaging system (AniView Phoenix full-spectrum

animal imaging system, China). Besides, the adhesion of nanoparticles on the thrombus surface was observed by SEM.

## In vivo safety study
To evaluate the in vivo safety of the polymersomes, mice were intravenously administered with CP@tPA at a dose equivalent to 1 mg kg⁻¹ of tPA. After 24 h, the mice were euthanized, and major organs (heart, liver, spleen, lungs, and kidneys) were harvested for histological analysis via H&E staining. Additionally, blood samples were collected and centrifuged at 700 × g for 5 min to obtain serum. The serum was subsequently used for blood biochemistry tests to assess liver and kidney function.

## In vivo thrombosis targeting[72]
MCAO mice were intravenously injected with Null@ICG, P@ICG, and CP@ICG polymersomes (400 µg polymersomes/mouse) at 2 h poststroke, and observed at different points by in vivo imaging systems. 2 h post injection, mice were executed to separate brain, heart, liver, spleen, lung and kidney for ex vivo imaging. The average radiation efficiency was analyzed using Phoenix software. Immunofluorescence staining was applied to analyze the thrombosis targeting of polymersomes. The brain tissue was cut into 20 µm sections and incubated with primary antibodies including rat anti-CD31 antibody (1:100, Abcam, ab256569) and rabbit anti-fibrinogen antibody (1:1000, Abcam, ab92572), and secondary antibodies including Alexa Fluor 594-conjugated donkey anti-rabbit IgG (1:1000, Abcam, ab150076) and Alexa Fluor 488-conjugated goat anti-rat IgG (1:1000, Abcam, ab150157), respectively. The nucleus was stained with DAPI. Images were obtained by confocal microscope (A1R +, Nikon, Japan).

## In vivo thrombolysis
The thrombolysis effect of CP@tPA in carotid artery thrombus model, lower extremity arterial thrombus model, and MCAO model was investigated. For both the carotid artery thrombus model and lower extremity arterial thrombus model, saline, tPA, or CP@tPA (150 µL per mouse, equivalent to 1 mg kg⁻¹ of tPA) was administered via tail-vein injection 10 min after model establishment to initiate thrombolytic therapy. For the MCAO model, 2 h after stroke onset the mice were intravenously given saline, tPA, or CP@tPA (150 µL per mouse, equivalent to 1 mg kg⁻¹ of tPA). Blood perfusion in the ischemic area was monitored by laser speckle at different time points (0, 30, 60, 90, and 120 min). For the lower extremity arterial thrombus model, mice were euthanized 2 h after thrombolysis to separate the blood vessels containing thrombosis for H&E staining. The thrombosis rate was calculated by ImageJ. Besides, thrombolysis was measured by observing the activated platelet in thrombus. Mice were injected with 100 µL of FITC-dextran (2 mg mL⁻¹, 2000 KDa) and 10 µL of APC-CD62P (Biolegend, 148303), and observed by two-photon microscopy (A1RMP +, Nikon, Japan). The thrombus markers FDP, sCD40L in mouse blood were determined by ELISA.

## Cerebral hemorrhage evaluation
At 22 h after treatment with tPA and CP@tPA (1 mg kg⁻¹ of tPA), MCAO mice were sacrificed to collect brains, which were sliced and photographed. The severity of hemorrhage was classified into four levels: no hemorrhage (NH); hemorrhage infarction type 1 (HI-I), a single petechia in the ischemic area; hemorrhage infarction type 2 (HI-II), 2 - 3 petechiae in the ischemic area; and hemorrhage infarction type 3 (HI-III), more than 3 petechiae in the ischemic area. The tissues were then homogenized and centrifuged, and the hemoglobin content was determined by measuring the absorbance at 410 nm using a hemoglobin assay kit. A standard curve was constructed with freshly collected homologous blood to calculate the total blood volume.

## Transcriptome analysis

Blood was collected from ischemic stroke patients with HT before and after tPA thrombolysis. The patients ranged in age from 67 to 91 years, including 2 males and 7 females. Clinical samples were obtained from the Third People Hospital of Chengdu (China). The collected blood was treated with Trizol to extract RNA. The transcriptome analysis was performed by constructing the libraries and sequencing by Shanghai Majorbio Bio-Pharm Biotechnology Co., Ltd. (China). The study was approved by the Ethics Review Committee of the Third People Hospital of Chengdu (2024-S-154), and informed consent signed by patients was obtained. The transcriptome analysis for the brain tissues was performed as the similar procedure except isolating the brain to be homogenized for RNA extraction.

To identify DEGs between groups, the expression level of each transcript was calculated according to the transcripts per million (TPM) method. Gene abundance estimation was performed with RSEM. Differential expression analysis was performed using DESeq2 (v1.24.0). DEGs with $|\log_2FC| \geq 1$ and FDR $\leq 0.05$(DESeg2) were considered to be significantly different expressed genes. Functional-enrichment analysis including Gene Ontology (GO), Reactome and KEGG were performed to identify biological functions and pathways significantly associated with the DEGs. A Benjamini-Hochberg corrected $P$-value $\leq 0.05$ was used as the threshold for significance against the whole-transcriptome background. GO enrichment was conducted using Goatools, while KOBAS was employed for both Reactome and KEGG pathway analyses.

## Effect of tPA on activating CD177+ neutrophils

The effect of tPA on the activation of CD177+ neutrophils was explored both in vitro and in vivo. For in vitro study, neutrophils were isolated from peripheral blood of MCAO mice and treated with tPA (100 μg mL$^{-1}$) for 90 min. Simply put, the mouse blood was mixed with PBS and red blood cell sedimentation solution at a volume ratio of 1:1:1. The mixture was left at room temperature for 30 min and centrifuged through density gradient to collect the supernatant and obtain the neutrophils. For in vivo study, tPA (1 mg kg$^{-1}$) was intravenously injected into MCAO mice at 2 h poststroke, and neutrophils were extracted from peripheral blood of mice with HT (petechia in the ischemic area) 22 h later. These cells were labeled with FITC-Ly6G (1:100, Thermo Fisher scientific, 11-9668-80) and APC-CD177 (1:50, BD, 566599) for flow cytometry (CytoFLEX, Beckman Coulter, USA) analysis. The percentage of Ly6G+CD177+ cells was calculated. Besides, brains were collected from MCAO mice with HT for immunofluorescence staining. Brain was cut into sections and incubated with primary antibodies including rat anti-Ly6G antibody (1:100, Abcam, ab25377) and rabbit anti-CD177 antibody (1:1000, Servicebio, GB11316-100), followed by treating with secondary antibodies including Alexa Fluor 488-conjugated goat anti-rat IgG (1:1000, Abcam, ab150157), and Alexa Fluor 594-conjugated donkey anti-rabbit IgG (1:1000, Abcam, ab150076). The nuclei were stained with DAPI. Images were obtained by confocal microscope.

## Effect of tPA on NET generation

Neutrophils were isolated from peripheral blood of MCAO mice and treated with tPA (100 μg mL$^{-1}$) for 90 min. Then the cells were rinsed, fixed with 4% paraformaldehyde at 4 °C, stained with SYTOX Green (1:500), and observed with fluorescence microscope. The concentration of NETs and MPO were measured by ELISA.

## CD177+ neutrophils promote BBB leakage

After 10 h of tPA thrombolysis, a cranial window was established to observe the BBB penetration of CD177+ neutrophils. Briefly, mice were anesthetized to fix the head on the brain locator. A circular area was drilled in the infarct brain with a micro handheld cranial drill and subsequently covered with a coverslip. Then mice were intravenously injected with FITC-dextran (10 mg kg$^{-1}$, 40 kDa) and APC-CD177

(0.1 mg kg$^{-1}$, BD, 566599), respectively. Images were acquired using two-photon microscopy at different time points.

## Effect of rCD177 on inhibiting CD177+ neutrophils migration

To explore the effect of rCD177 on inhibiting the transendothelial ability of CD177+ neutrophils, transmigration assay was performed[38]. bEnd.3 were seeded on the upper chamber of transwell and cultured to form cell monolayer, which could be used as the BBB model once the value of trans endothelial electrical resistance (TEER) was higher than 200 Ω· cm². The culture medium was then replaced with glucose-free and serum-free DMEM medium (Gibco), and the cells were cultured in 95% N₂ and 5% CO₂ environment to simulate low oxygen condition[73]. After 3 h of oxygen-glucose deprivation (OGD) processing, the medium was replaced with normal medium and the cells were transferred to normal incubator to simulate reperfusion injury. rCD177 (100 μg mL$^{-1}$) was added to the upper chamber for 10 min, subsequently neutrophils ($1 \times 10^6$) and tPA (100 μg mL$^{-1}$) were added for further incubation. 90 min later, cells were collected from the lower chamber, and labeled with FITC-Ly6G (1:100, Thermo Fisher scientific, 11-9668-80) and APC-CD177 (1:50, BD, 566599) for flow cytometry analysis. Besides, the number of neutrophils in the lower chamber was counted to calculate cell mobility. Then the migrated cells were stained with SYTOX Green (1:500) and observed with fluorescence microscope. The concentration of NETs was measured by ELISA kit.

## Inhibition of CD177+ neutrophils migrating into the brain

rCD177 was fluorescently labeled with FITC and encapsulated into CP polymersomes. The resulting CP@rCD177 was administered to MCAO mice 2 h poststroke and 10 min prior to thrombolysis. At 22 h posttreatment, brain tissues were cut into 20 μm sections and subjected to immunostaining using primary antibodies including rat anti-CD31 antibody (1:100, Abcam, ab256569) and rabbit anti-CD177 antibody (1:1000, Servicebio, GB11316-100), followed by incubation with secondary antibodies including Alexa Fluor 594-conjugated donkey anti-rabbit IgG (1:1000, Abcam, ab150076) and Alexa Fluor 647-conjugated goat anti-rat IgG (1:1000, Abcam, ab150159). Nuclei were stained with DAPI. Images were obtained by confocal microscope.

For further investigation, mouse brains were collected from saline, tPA, rCD177+tPA, CP@tPA, and CP@rCD177/tPA groups after treatment. The tissues were homogenized and labeled with FITC-Ly6G (1:100, Thermo Fisher scientific, 11-9668-80) and APC-CD177 (1:50, BD, 566599) for flow cytometry analysis. For immunofluorescence staining, brains were cut into sections and incubated with primary antibodies including rat anti-Ly6G antibody (1:100, Abcam, ab25377) and rabbit anti-CD177 antibody (1:1000, Servicebio, GB11316-100), followed by treating with secondary antibodies including Alexa Fluor 488-conjugated goat anti-rat IgG (1:1000, Abcam, ab150157) and Alexa Fluor 594-conjugated donkey anti-rabbit IgG (1:1000, Abcam, ab150076). Images were obtained by confocal microscope.

## Effect of CP@rCD177 on improving tPA-induced HT

MCAO mice were intravenously injected with saline, tPA, rCD177/tPA, CP@tPA, and CP@rCD177/tPA at 2 h poststroke (1 mg kg$^{-1}$ of tPA and 1 mg kg$^{-1}$ rCD177). For rCD177/tPA and CP@rCD177/tPA groups, rCD177 and CP@rCD177 were injected 10 min prior to tPA and CP@tPA injections, respectively. 22 h after therapy, the blood leakage in the mouse brains was detected using the photoacoustic multimode small animal in vivo imaging system (GAni-Plus, Guangzhou G-Cell Technology Co., Ltd.). Mice were anesthetized and the skin tissue of the mice's head was cut open to expose the skull, which was infiltrated with deionized water to couple the photoacoustic signals. The hemoglobin fluorescence was detected at 532 nm to reconstruct the hemoglobin signal. Then the mouse brains were collected, sliced and photographed. The severity of hemorrhage was evaluated by four levels.

Additionally, the brain tissue was homogenized and centrifuged; the hemoglobin content was then quantified by measuring the absorbance at 410 nm using a hemoglobin assay kit. A standard curve was generated with freshly collected homologous blood to calculate the total hemorrhage volume.

## EB staining
Mice were intravenously injected with 4% EB (4 mL kg$^{-1}$) 22 h after treatment. 2 h later, mice were executed and brains were cut into 1 mm slices for photographing by camera. Then the tissues were homogenized and centrifuged to collect the supernatant. The EB content was determined by measuring the absorbance at 620 nm, expressed as $A_{620}$/tissue (g).

## Assessment of BBB integrity
BBB integrity was first assessed in vivo using IntraVital microscopy (IVM-CMS-3, Shanghai Aifei Electronic Technology Co., Ltd). At 22 h post-treatment, a cranial window was implanted over the infarcted hemisphere of MCAO mice, followed by intravenous injection of 100 μL Evans blue (1 mg mL$^{-1}$). After 30 min, intravital imaging was performed, revealing extravasation of the dye from compromised cerebral vessels. In parallel, TEM was employed to examine the ultrastructure of the BBB at 22 h post-treatment. Brain tissue was collected, fixed in 2.5% glutaraldehyde, dehydrated, embedded, and sectioned into ultrathin slices. The sections were stained with uranyl acetate and lead citrate, and the microarchitecture of the BBB was visualized and imaged under TEM.

## TTC staining
In total, 22 h after treatment, the mouse infarcted brains were harvested, and sectioned coronally into five 1 mm-thick slices. The slices were incubated in a 2% TTC solution at 37 °C for 30 min, fixed in 4% paraformaldehyde, rinsed with PBS, and imaged using a digital microscope. Viable brain tissue was stained rose-red, whereas the infarcted areas remained unstained (white). The total area and the infarct area of each slice were quantified using ImageJ software.

$$\text{Infarct area}(\%) = \frac{\text{Todal erbral infartction area}}{\text{Total brain area}} \times 100\% \qquad (6)$$

## Inflammatory cytokine assay
To evaluate neuroinflammatory responses after treatment, mice were euthanized on day 3 post-treatment. The ischemic brain hemispheres were collected, and the concentrations of key pro-inflammatory cytokines (IL-1β and TNF-α) as well as anti-inflammatory cytokines (IL-10 and TGF-β) were quantified using ELISA kits[74].

## Behavioral test
mNSS. Mice underwent comprehensive behavioral assessments to evaluate their motor, sensory, reflexive, and equilibrium functions for the mNSS calculation. The mNSS scale ranges from 0 to 18 points, where 0 indicates no observable neurological deficits and 18 represents the most severe neurological impairment.

Corner turning test. The corner turning test was employed to assess behavioral bias in mice. Two plastic plates were arranged at a 30° angle with a 5 mm gap at the vertex to create a corner that attracted the mice. Each mouse was allowed to enter the corner and then spontaneously turn either left or right. The test was repeated ten times per mouse, and the number of left and right turns was recorded. The turning bias was calculated using the following formula:

$$\text{Corner turn score}(\%) = \frac{R}{L + R} \times 100\% \qquad (7)$$

L: left turn, R: right turn.

Rotarod test. The motor coordination of mice was assessed using a rotarod test by Rota Rod System (LE8200, RWD). Mice were placed on a rotating rod that accelerated gradually from 5 to 40 rpm over a 4-min period. Before formal testing, all mice received a 5-day training regimen consisting of 3 trials per day. Mice that failed to remain on the rod for 270 s during training were excluded from subsequent experiments. The latency to fall was recorded and analyzed as the mean residence time.

Adhesive test. Adhesive test was used to evaluate cutaneous sensitivity and sensorimotor integration. Specifically, a piece of adhesive paper (3 × 3 mm$^2$) was applied to the ipsilateral forepaw of each mouse, and the time taken for the mouse to contact and remove the paper was recorded. Similarly, mice were trained before surgery to exclude those that could not remove the paper after 60 s.

Barnes maze test[75]. Barnes maze test was used to test spatial cognitive function of mice after stroke. Briefly, mice were placed in the center of the maze, which was covered with an impervious box to limit their activity to 5 s. Then, the box was removed to expose the mice to strong light above the platform, forcing them to search for the correct hole connected to the dark box. Before MCAO surgery, the mice were trained for five consecutive days (three trials per day) to acquire the spatial memory required to locate the target escape hole, and any mouse that failed to escape within 5 min was excluded. The whole process of testing was videotaped camera until the mice entered the escape hole. The movement route was analyzed using the SMART 3.0 behavioral recording system (Panlab, Spain).

## Western blot
Protein expression levels were examined through western blot analysis. Brain tissues were collected and lysed in RIPA buffer. The lysate was then detected using BCA protein analysis kit to quantify the protein content. Equal amount of protein was isolated by sodium dodecyl sulfate-polyacrylamide gel electrophoresis (SDS-PAGE), and transferred to the nitrocellulose membrane. At room temperature, the membrane was closed by 5% skim milk powder for 2 h, washed with TBST, and incubated with rabbit anti-GADD45 (1:1000, Thermo Fisher scientific, PA5-43160), rabbit anti-TSP1 (1:1000, Solarbio, K007665P), rabbit anti-ANGPT2 (1:1000, Solarbio, K001733P), rabbit anti-COX-2 (1:500, Solarbio, K009752P), and anti-β-actin (1:1000, Cell Signaling Technology, 4967S) overnight. Next, the sample was washed with TBST, followed by incubating with horseradish peroxidase-conjugated secondary antibody. Protein strips were imaged with ECL development, the light density of protein strips was quantified. β-actin was served as loading control.

## In vivo NET decomposition
To observe the in vivo NET decomposition, mice after treatment with different formulations were intravenously injected with SYTOX Green (10 μM), PE-Ly6G (10 μL, BioLegend, 127607) and EB (1 mg mL$^{-1}$, 100 μL). 30 min later, mice were sacrificed to separate the brains and observed by IntraVital Microscopy. Besides, the brains were homogenized and centrifuged to obtain the supernatant. The concentrations of NETs and inflammatory factors (IL-6, IFN-β) were measured by ELISA.

## Brain distribution of NETs
HT mice were executed to isolate the brains 24 h poststroke. The brains were cut into sections and incubated with primary antibodies including rat anti-Iba1 antibody (1:100, Abcam, ab283346), rat anti-GFAP antibody (1:200, Abcam, ab279291), or rat anti-NeuN antibody (1:100, Abcam, ab279297), and rabbit anti-H3Cit (1:1000, Abcam, ab219407). Then the brain sections were treated with secondary antibodies including Alexa Fluor 488-conjugated goat anti-rat IgG (1:1000, Abcam, ab150157) and Alexa Fluor 594-conjugated donkey anti-rabbit

IgG (1:1000, Abcam, ab150076), and visualized using confocal microscope.

## Statistical analysis

Statistical analysis was performed using GraphPad Prism (9.5.0) software. For comparison of means of samples with normal distribution and homogeneous variance, paired *t* test was used for two groups. For samples with more than two groups, one-way or two-way ANOVA with Bonferroni's or Tukey's post hoc test was used for multiple comparisons. All data are presented as means ± SDs. $*P < 0.05$, $** P < 0.01$, $*** P < 0.001$, $**** P < 0.0001$, n.s, not significant.

## Reporting summary

Further information on research design is available in the Nature Portfolio Reporting Summary linked to this article.

## Data availability

All data supporting the results of this study are included in the article and Supplementary Information. The RNA-seq has been deposited to NCBI under accession codes (PRJNA1331951, PRJNA1332825). Source data are provided with this paper.

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

## Acknowledgements

This work was supported by the National Natural Science Foundation of China (U22A20161, 52495011 to S.Z, 52473324 to X.G), the National Key R&D Program of China (2022YFA1206500 to X.G), and Sichuan Science and Technology Program (2024NSFTD0002 to X.G). The authors thank Analytical and Testing Center of Southwest Jiaotong University.

## Author contributions

Z.W and X.G conceived and designed the project. Z.W, Z.X, and C.H performed experiments and data analysis. X.G, H. L., and S.Z. supervised the research. All authors participated in drafting the manuscript, engaging in discussions on the results and their implications, and revising the paper throughout its development.

## Competing interests

The authors declare no competing interests.
