## [Transparent Peer Review file · Nature Communications]

Polymersomes preventing brain infiltration of CD177⁺ neutrophils to mitigate hemorrhagic transformation post-tPA thrombolysis

Corresponding Author: Professor Xing Guo

This file contains all reviewer reports in order by version, followed by all author rebuttals in order by version. Unpublished, confidential data in this file is redacted.

Version 0:

Reviewer comments:

Reviewer #1

(Remarks to the Author)

The authors have developed a thrombus-targeting and ROS-triggered polymersomes to separately load tPA and recombinant CD177 protein (rCD177) for thrombolysis while concurrently inhibiting hemorrhagic transformation. Overall, this manuscript is well organized. There are some issues may be further improved before publication:

1. The characterization of the CP@rCD177 nanoparticles is incomplete, it is recommended to add experiments to prove the successful loading of the drug.
2. In the drug release experiment (Figure 1d), less than 25% of the drug was released within 2 h. Why is the thrombolytic effect at 2 h stronger than tPA (Figure 2b)? It is recommended to add an explanation of why thrombolysis can be achieved without drug release.
3. The colloidal stability (FBS) and long-term particle stability should be provided for in vivo applications and recent reference, such as Nat. Commun. 14, 255 (2023), Angew. Chem. Int. Ed. 61, e202109068 (2022) would be helpful for explanation and discussion.
4. It is recommended to supplement the determination method of the dosing interval and explain how sequential administration ensures the drug's duration of action.
5. The bleeding risk was determined in the MCAO model. It is recommended to further verify the bleeding risk in other models to enhance the generalizability of the findings. Additionally, it is suggested to conduct safety evaluations of blood cells and vascular endothelial cells after treatment.
6. Unable detect the process of CP@rCD177 nanoparticles in vivo. The pharmacokinetics studies should be provided.
7. The administration method is unreasonable. Clinically, it is impossible to administer CP@rCD177 before the onset of the disease. The authors should attempt to administer the drug simultaneously with model establishment.
8. Reperfusion injury caused by thrombolysis can easily trigger various cerebral inflammations. It is recommended to measure other inflammatory factors in the brain after treatment. Otherwise, it will be impossible to determine the inflammatory level in the brain and the treatment effect after treatment.
9. The overall standardization of the manuscript is subpar.
 - a) The content of sub-figure e-f in Fig. 1 does not match the figure title.
 - b) Fig. 4f is a flow cytometry image, which is inconsistent with the description in the article stating that "Photoacoustic imaging (PA) was used for real-time monitoring of intracerebral hemorrhage.
 - c) The content of sub-figures b-c in Fig. 2 is chaotic.

Reviewer #2

(Remarks to the Author)

The study by Wang et al. presents a new therapeutic platform to deliver thrombolysis after ischemic stroke in mice with diminished risk of hemorrhagic transformation (HT). The intervention is based on synthetic particles that are targeted to the thrombus by a fibrin-binding peptide and contain encapsulated rtPA. Furthermore, the authors loaded the particles with recombinant CD177 to block neutrophil-endothelial cell interaction which resulted in reduced BBB breakdown and neutrophil migration to the stroked brain. Combinatorial nanoparticles incorporating tPA and rCD177 proved more potent in preventing HT. This is a well-designed study that shows the efficacy of a novel ischemic stroke therapy with potential for clinical translation. However, the quantitative aspects of the study are limited, data are not well presented, and the benefits

for stroke outcome have not been investigated. Specific concerns are:

1. There is no indication that CP@tPA reduced infarct volume and improved neurobehavioral outcome after stroke. These are essential – and the only clinically relevant – parameters for any therapeutic intervention.
2. It is presumed that 100 μM H_2O_2 was used for all in vitro experiments. The concentrations of extracellular H_2O_2 released by neutrophils is thought to be in the low μM range (PMID: 6323407). In addition, the presence of erythrocytes in the clot would act as a H_2O_2 sponge further reducing available H_2O_2 for polymer breakdown.
3. Ferric chloride clots are relative resistant to tPA thrombolysis because of low fibrin incorporation into the clot (PMID: 21326267). What was the recanalization rate in tPA and CP@tPA treated FeCl_3 MCAO mice? Why would CP@tPA be superior in dissolving a clot that has minimal fibrin deposition?
4. Fig 2h. The HT in the CP@tPA coronal sections looks different between tPA and CP@tPA treated animals. While the number of macroscopic petechial bleeds might be higher in tPA treated mice, there is indication of a more diffuse blood extravasation in CP@tPA mice. At least this is my impression from the presented images. In addition, petechial bleeds are also observed in the contralateral hemisphere of CP@tPA mice. Contralateral and ipsilateral hemispheres of all treatment groups should be used to quantitatively determine the hemoglobin content by chemical analysis.
5. Fig S9 shows extensive accumulation of polymersomes in lung, kidney, and liver. How and when are polymersomes cleared from those organs and what is their effect on lung, kidney, and liver tissue integrity and function?
6. Many experiments are missing time points of drug treatment and endpoints. Generally, time points should be indicated in the main text and/or figure legends, and not in the methods section.
7. The SYTOX indicator should be used in conjunction with a neutrophil marker such as Ly6G.

Minor

1. In what volume were the polymersomes and tPA delivered?
2. There are no clinical data for stroke patients and controls.
3. Methods for RNA-seq and DGE, including statistical analysis, are missing.
4. Line 504. Please specify “lower extremity artery”.
5. Fig 1e-f. Reversed order in legend.
6. Fig S7a. Size markers missing.

Version 1:

Reviewer comments:

Reviewer #1

(Remarks to the Author)

The authors well address my concerns.

Reviewer #2

(Remarks to the Author)

The authors have added relevant clarifications and provided new data and experiments to support their findings, thereby improving the overall quality of the study. One concern remains, that needs to be addressed. While there is an overall decrease in petechiae and larger bleedings in CP@tPA treated animals, the total amount of brain hemoglobin was not reduced (Fig. R4). Furthermore, the authors refer to Fig. 4J as quantitative hemoglobin measurements (“hemoglobin detection kit”) while the graph seems to depict hemoglobin volume obtained by morphometric analysis. This discrepancy needs to be addressed.

In addition, the presence of diffuse hemoglobin signals in CP@tPA might reflect microvascular problems, as acknowledged by the authors in their rebuttal, and might be an important factor for the clinical translation of this therapeutic approach. It needs, therefore, further investigation (high resolution histology?), and Fig R4 should be presented as supplemental data. The authors are also encouraged to include these findings into the discussion.

Version 2:

Reviewer comments:

Reviewer #2

(Remarks to the Author)

The authors have resolved the remaining concerns by adding further experiments and analyses, and the revised discussion now more clearly reflects the complex aspects of thrombolytic therapies for ischemic stroke.

Point-by-point responses to Reviewers' Comments

We would like to express our appreciation for the thoughtful critiques provided by Reviewer and we have used as the basis for an extensive revision of the manuscript. The reviewers' comments are provided in italics. Our response to each comment is provided in plain text.

With Regard to the Comments of Reviewer #1:

Reviewer #1: The authors have developed a thrombus-targeting and ROS-triggered polymersomes to separately load tPA and recombinant CD177 protein (rCD177) for thrombolysis while concurrently inhibiting hemorrhagic transformation. Overall, this manuscript is well organized. There are some issues may be further improved before publication.

Re: We sincerely appreciate the reviewer's thorough review and encouraging comments. In response to the constructive suggestions aimed at further improving our work, we have carefully addressed each of the raised issues in the revised version of the manuscript. Below, we provide detailed responses to each comment point-by-point.

1. The characterization of the CP@rCD177 nanoparticles is incomplete, it is recommended to add experiments to prove the successful loading of the drug.

Re: Thank you for this helpful suggestion. In response, we have supplemented the characterizations of CP@rCD177 polymersomes including size distribution, surface potential, and drug loading. Dynamic light scattering (DLS) measurements revealed that the nanoparticles have a hydrodynamic diameter of 124.3 nm and a zeta potential of -3.85 ± 0.75 mV (**Supplementary Table 3**). Furthermore, the drug loading content and encapsulation efficiency of CP@rCD177 were determined to be 6.62% and 72.82%,

respectively, confirming the successful and efficient loading of rCD177.

Supplementary Table 3. Characterization of CP@rCD177 polymersomes ($n = 3$).

	Size	PDI	Zeta potential	Loading content (%)	Loading efficiency (%)
CP@rCD177	124.3 ± 2.77	0.183 ± 0.026	-3.85 ± 0.75	6.62 ± 0.13	72.82 ± 1.39

2. In the drug release experiment (Figure 1d), less than 25% of the drug was released within 2 h. Why is the thrombolytic effect at 2 h stronger than tPA (Figure 2b)? It is recommended to add an explanation of why thrombolysis can be achieved without drug release.

Re: We thank the reviewer for raising this important point. The apparent discrepancy between the drug release profile and the thrombolytic efficacy can be explained by the greatly extended circulation time of CP@tPA compared to free tPA. As shown in **Supplementary Table 2** and **Supplementary Fig. 9**, the half-life of free tPA is only approximately 6.99 min, meaning it is rapidly cleared from the bloodstream post-injection, significantly limiting its therapeutic window and cumulative thrombolytic activity. In contrast, CP@tPA exhibits a 9.41-fold longer half-life (65.75 min), allowing it to remain in circulation significantly longer.

Although less than 25% of the loaded drug is released within the first 2 h (**Fig. 1d**), the polymersome encapsulation provides sustained release over a prolonged period. Moreover, the incorporation of the CREKA peptide enables thrombus-targeting ability of polymersomes, facilitating selective accumulation of CP@tPA at the thrombus site. This ensures that a therapeutically effective concentration of tPA is maintained for an extended duration, resulting in more continuous and effective thrombolysis. Consequently, by the 2-h time point, the cumulative thrombolytic effect of CP@tPA exceeds that of free tPA (**Fig. 2b**), which has been largely cleared from the circulation. In essence, the enhanced efficacy is attributable to the improved pharmacokinetics and active targeting capability of CP polymersomes, rather than rapid drug release.

Supplementary Table 2. Pharmacokinetic parameters of tPA and CP@tPA ($n = 3$).

Sample	$t_{1/2}$	Ke (1/min)	V1 (L/kg)	CL (mL/min/kg)	AUC (0-t) (mg/L*min)
tPA	6.986±1.535	0.103±0.025	0.197±0.016	20.033±3.556	45.984±8.401
CP@tPA	65.747±3.750	0.010±0.001	0.217±0.010	2.285±0.069	287.025±10.846

Supplementary Fig. 9. *In vivo* pharmacokinetics profiles of tPA and CP@tPA ($n = 3$).

Fig. 1d. *In vitro* tPA release from CP polymersomes in PBS or H₂O₂ ($n = 3$).

Fig. 2b. Thrombolysis photographs of tPA and CP@tPA in lower extremity arterial

thrombosis model.

3. The colloidal stability (FBS) and long-term particle stability should be provided for *in vivo* applications and recent reference, such as *Nat. Commun.* 14, 255 (2023), *Angew. Chem. Int. Ed.* 61, e202109068 (2022) would be helpful for explanation and discussion.

Re: We sincerely appreciate the reviewer's insightful comments regarding the importance of colloidal and long-term stability for *in vivo* applications. In response, we have conducted systematic stability evaluations of both CP@tPA and CP@rCD177 polymersomes, in accordance with the suggested references.^{1,2} As shown in **Supplementary Fig. 6a** and **25a**, both formulations exhibited excellent colloidal stability over 8 h in FBS, with no significant changes in particle size or polydispersity index. Moreover, long-term stability assessment demonstrated that the polymersomes maintained structural integrity at 4 °C under dark conditions without considerable size variation for up to 30 days (**Supplementary Fig. 6b** and **25b**). We attribute this high stability primarily to the surface-grafted PEG chains, which effectively suppress nonspecific protein adsorption and minimize particle aggregation under physiological conditions. Collectively, these results confirm that our polymersome platform exhibits excellent colloidal and storage stability, fulfilling key requirements for *in vivo* translational applications.

Supplementary Fig. 6. a, Colloidal stability of CP@tPA incubated in FBS (pH 7.4) within 8 h at 37 °C ($n = 3$). **b**, Size distribution of CP@tPA in PBS for 30 days.

Supplementary Fig. 25. a, Colloidal stability of CP@rCD77 incubated in FBS (pH 7.4) within 8 h at 37 °C ($n = 3$). **b,** Size distribution of CP@rCD77 in PBS for 30 days.

References:

1. Zhang, H. et al. Molecularly self-fueled nano-penetrator for nonpharmaceutical treatment of thrombosis and ischemic stroke. *Nat. Commun.* **14**, 255 (2023).
2. Zhang, H. et al. NIR-II hydrogen-bonded organic frameworks (HOFs) used for target-specific amyloid- β photooxygenation in an alzheimer's disease model. *Angew. Chem. Int. Ed.* **61**, e202109068 (2022).

4. *It is recommended to supplement the determination method of the dosing interval and explain how sequential administration ensures the drug's duration of action.*

Re: Thank you for this important question regarding the rationale behind our dosing interval and the mechanism for sustained drug action. The selection of the 10-min pre-administration interval for CP@rCD177 before thrombolysis was based on a systematic evaluation of intravascular rCD177 distribution and its relationship to hemorrhagic transformation (HT) outcomes. As shown in our supplementary data (**Fig. R1a**), significant co-localization of rCD177 with CD31 in cerebral vessels was observed as early as 10 min after CP@rCD177 administration, indicating rapid target engagement at vascular sites. This prompt localization is crucial for modulating CD177⁺ neutrophil-mediated BBB disruption during the critical early phase of ischemia-reperfusion.

Interestingly, while extended intervals allowed greater rCD177 accumulation, they

did not translate into improved safety outcomes. Instead, a 10-min interval resulted in a significantly smaller hemorrhagic volume compared to a 30-min interval (**Fig. R1b**). We attribute this to the critical importance of timely reperfusion: delaying thrombolysis exacerbates ischemic injury and increases HT risk, as supported by previous studies.¹ Thus, the 10-min window represents an optimal balance between sufficient target engagement and timely restoration of blood flow.²

To ensure sustained pharmacologic activity throughout the acute phase, CP@rCD177 was formulated within polymersomes, which significantly prolong the plasma half-life of rCD177 (**Supplementary Fig. 25e** and **Table 4**). This delivery system enables durable vascular binding, as demonstrated by the persistent presence of rCD177 in cerebral vessels even 24 h after MCAO (**Fig. 4e**), thereby maintaining continuous suppression of CD177⁺ neutrophil-mediated vascular injury. In summary, our dosing strategy integrates both temporal optimization for maximal safety and a sustained-release formulation to ensure prolonged therapeutic effect.

Fig. R1. The impact of dosing intervals on hemorrhagic transformation outcomes.

a, Temporal colocalization of rCD177-FITC with CD31 in cerebral vessels after CP@rCD177-FITC injection. **(b)** Representative brain sections and **(c)** quantified hemorrhage volume at 22 h after thrombolysis with CP@tPA administered at different time points ($n = 10$). Data are presented as the mean \pm s.d. P values were calculated by one-way ANOVA followed by Tukey's post-hoc test.

Reference:

1. García-Culebras, A. et al. Toll-like receptor 4 mediates hemorrhagic transformation after delayed tissue plasminogen activator administration in in situ thromboembolic stroke. *Stroke* **48**, 1695-1699 (2017).
2. Wang, R. et al. Neutrophil extracellular traps promote tPA-induced brain hemorrhage via cGAS in mice with stroke. *Blood* **138**, 91-103 (2012).

5. The bleeding risk was determined in the MCAO model. It is recommended to further verify the bleeding risk in other models to enhance the generalizability of the findings. Additionally, it is suggested to conduct safety evaluations of blood cells and vascular endothelial cells after treatment.

Re: Thank you for this important suggestion. In response to your comment, we would like to clarify the rationale behind focusing on the MCAO model for bleeding risk assessment. Hemorrhagic transformation (HT) typically occurs under specific physiological conditions: namely, the presence of richly vascularized tissues with underlying vessel injury, along with soft surrounding tissue.^{1,2} These conditions are characteristically present in the brain during ischemic stroke, making HT a clinically significant and potentially life-threatening complication of thrombolytic therapy.^{3,4} In contrast, vessels in regions such as the lower extremity and carotid artery are generally surrounded by thicker walls and denser connective tissue, which inherently reduces the risk of severe hemorrhage. In our experiments, no hemorrhage was observed in either the carotid artery thrombus model or lower extremity arterial thrombus model at 22 h after thrombolysis (**Fig. R2**), supporting the view that HT in these models is of limited clinical relevance. Despite this, we agree that extending this research to other models would be valuable. Investigating the broader relevance of this protective effect across different vascular beds will be a critical next step in definitively establishing the generalizability of the treatment.

Fig. R2. Digital photos showing tissues in (a) carotid artery thrombus and (b) lower extremity arterial thrombus mouse models 22 h after thrombolysis.

Regarding the safety evaluation of blood cells and vascular endothelial cells post-treatment, we conducted a series of *in vitro* experiments including cytocompatibility and hemocompatibility evaluations. In terms of cytocompatibility, when CP@tPA was co-cultured with bEnd.3 cells, cell viability remained consistently above 85% (**Supplementary Fig. 7**), indicating no significant cell toxicity. In terms of hemocompatibility, the hemolysis rate of CP@tPA was below 5%, meeting the safety standards for biomaterials, and the blood clotting index (BCI) was comparable to that of the saline control group (**Supplementary Fig. 8**). This demonstrates that the formulation does not abnormally activate or interfere with the coagulation system, confirming its favorable hemocompatibility.

Supplementary Fig. 7. (a) Live staining fluorescence (b) and cell viability showing *in vitro* cytotoxicity of bEnd.3 treated with different concentrations of CP@tPA polymersomes ($n = 5$).

Supplementary Fig. 8. (a) Hemolytic ratio (b) and hemolysis photographs of CP@tPA polymersomes at different concentrations ($n = 3$). c, d) Evaluation of total blood coagulation with different concentrations of CP@tPA polymersomes ($n = 3$).

Reference:

1. Bergs, J. et al. The Networking Brain: How Extracellular Matrix, Cellular Networks, and Vasculature Shape the In Vivo Mechanical Properties of the Brain. *Adv. Sci.* **11**, e2402338 (2024).
2. Bernardo-Castro, S. et al. Pathophysiology of Blood-Brain Barrier Permeability Throughout the Different Stages of Ischemic Stroke and Its Implication on Hemorrhagic Transformation and Recovery. *Front. Neurol.* **11**, 594672 (2020).
3. Shi, K. et al. tPA mobilizes immune cells that exacerbate hemorrhagic transformation in stroke. *Circ. Res.* **128**, 62-75 (2021).
4. Zhang, H. et al. Molecularly self-fueled nano-penetrator for nonpharmaceutical treatment of thrombosis and ischemic stroke. *Nat. Commun.* **14**, 255 (2023).
6. *Unable detect the process of CP@rCD177 nanoparticles in vivo. The pharmacokinetics studies should be provided.*

Re: Thank you for this important suggestion. We have performed pharmacokinetic studies to evaluate the *in vivo* behavior of CP@rCD177 nanoparticles. The results demonstrate that, compared to free rCD177, CP polymersomes significantly extend the half-life of rCD177 from 17.15 min to 66.6 min (**Supplementary Fig. 25** and **Supplementary Table 4**). This prolonged circulation time enhances the opportunity of rCD177 for sustained interaction with CD31 on cerebral vessels, thereby facilitating more effective inhibition of CD177⁺ neutrophil transmigration. The complete pharmacokinetic data have been included in the revised manuscript.

Supplementary Fig. 26. e, *In vivo* pharmacokinetic profiles of CP@rCD177 and rCD177 ($n = 3$).

Supplementary Table 4. Pharmacokinetic parameters of rCD177 and CP@rCD177 ($n = 3$).

Sample	$t_{1/2}$	Ke (1/min)	V1 (L/kg)	CL (mL/min/kg)	AUC (0-t) (mg/L*min)
rCD177	17.146±2.366	0.041±0.006	0.248±0.020	10.103±1.061	83.375±3.577
CP@rCD177	66.602±2.319	0.010±0.001	0.212±0.005	2.208±0.094	291.585±1.320

7. *The administration method is unreasonable. Clinically, it is impossible to administer CP@rCD177 before the onset of the disease. The authors should attempt to administer the drug simultaneously with model establishment.*

Re: We thank the reviewer for raising this important point regarding the clinical relevance of our administration protocol. We fully agree that pretreatment before disease onset is not clinically feasible. In our study, CP@rCD177 was administered 2 h after MCAO induction and 10 min before tPA thrombolysis. This schedule was designed to closely mirror the clinical scenario where patients receive treatment only after arriving at the hospital post-stroke, and thrombolytic agents are administered following diagnosis. While simultaneous administration with model establishment could offer mechanistic insights, it does not align with the actual clinical situation, since stroke patients are unable to receive immediate intervention at the exact onset of the event. In reality, there is typically a delay of several hours due to the time required for patient transportation, hospital arrival, diagnostic evaluation, and initiation of treatment.

Despite this, in future studies, we will include additional experimental groups with administration at the time of model establishment to further dissect the underlying biology. To improve clarity, we have included a schematic diagram explicitly illustrating the timeline of MCAO model establishment and drug administration (**Fig. 4d**), which we hope will help avoid further misunderstanding. We appreciate the reviewer's constructive comments, which have helped us better contextualize our experimental design and strengthen the clinical relevance of our study.

Fig. 4d. Schematic illustration of CP@rCD177 intervention before thrombolysis. CP@rCD177 was administered at 2 h after MCAO, followed by CP@tPA injection 10 min later. At 24 h poststroke, CD177⁺ neutrophil expression and hemorrhagic transformation were investigated.

8. Reperfusion injury caused by thrombolysis can easily trigger various cerebral

inflammations. It is recommended to measure other inflammatory factors in the brain after treatment. Otherwise, it will be impossible to determine the inflammatory level in the brain and the treatment effect after treatment.

Re: Thank you for this valuable suggestion. We agree that assessing a broader panel of inflammatory markers is important to comprehensively evaluate cerebral inflammation following thrombolysis and reperfusion injury. To assess post-treatment neuroinflammation, we supplemented ELISA measurement of key pro-inflammatory cytokines (IL-1 β , TNF- α) and anti-inflammatory cytokines (IL-10, TGF- β) in the ischemic hemisphere on day 3 after treatment. As shown in **Supplementary Fig. 31**, both cerebral ischemia (MCAO group) and hemorrhagic transformation (tPA and CP@tPA groups) resulted in significantly elevated pro-inflammatory cytokines and reduced anti-inflammatory mediators. This inflammatory response is likely triggered by ischemia-reperfusion injury, tPA-induced neutrophil chemotaxis, and microglial activation, as supported by previous studies.¹⁻⁴ Importantly, CP@rCD177 therapy significantly attenuated this response, reducing IL-1 β and TNF- α levels by 12.1% and 48.5%, respectively, while increasing IL-10 and TGF- β by 28.6% and 39.6%, respectively, compared to the MCAO group. These results suggest that targeting CD177⁺ neutrophils can effectively alleviate thrombolysis-related neuroinflammation. Thank you again for this constructive comment, which has helped us improve the depth of our analysis.

Supplementary Fig. 31. Expression levels of pro-inflammatory cytokines (IL-1 β , TNF- α) and anti-inflammatory cytokine (IL-10, TGF- β) in the ischemic brains of

MCAO mice on day 3 poststroke. Data are presented as the mean \pm s.d. *P* values were calculated by one-way ANOVA followed by Tukey's post-hoc test.

Reference:

1. Xiao, X. et al. A "Nano-Courier" for Precise delivery of acetylcholine and melatonin by C5a-Targeted aptamers effectively attenuates reperfusion injury of ischemic stroke. *Adv. Funct. Mater.* **33**, 2213633 (2023).
2. Zhang, M. et al. Ischemia-reperfusion injury: molecular mechanisms and therapeutic targets. *Signal Transduct. Target. Ther.* **9**, 12 (2024).
3. Yin, N. et al. A neutrophil hijacking nanoplatfom reprograming NETosis for targeted microglia polarizing mediated ischemic stroke treatment. *Adv. Sci.* **11**, e2305877 (2024).
4. Wang, R. et al. Neutrophil extracellular traps promote tPA-induced brain hemorrhage via cGAS in mice with stroke. *Blood.* **138**, 91-103 (2021).

9. The overall standardization of the manuscript is subpar. a) The content of sub-figure e-f in Fig. 1 does not match the figure title. b) Fig. 4f is a flow cytometry image, which is inconsistent with the description in the article stating that "Photoacoustic imaging (PA) was used for real - time monitoring of intracerebral hemorrhage. c) The content of sub -figures b-c in Fig. 2 is chaotic.

Re: We sincerely apologize for the oversights and errors in the figure presentation and labeling that led to confusion. We greatly appreciate the reviewer's careful reading and valuable comments. We have carefully corrected all the issues mentioned:

a) Regarding **Fig. 1e** and **1f**, the captions were inadvertently swapped. We have now corrected the titles to accurately match the content of each subfigure.

b) In **Fig. 4f**, the panel is indeed a flow cytometry image and was mistakenly referred to as a photoacoustic imaging result in the text. We have corrected this error by updating the corresponding labelling in the manuscript to accurately reflect that **Fig. 4f** represents flow cytometry data.

c) For **Fig. 2b** and **2c**, the previous version incorrectly referred to a carotid thrombus

model in **Fig. 2b**. Both subfigures actually illustrate the thrombolysis effect in a lower extremity arterial thrombosis model. The labels and descriptions have been revised to reflect the correct model.

All these corrections have been carefully implemented in the revised manuscript. Thank you once again for highlighting these issues, which have significantly improved the clarity and accuracy of our figures and text.

With Regard to the Comments of Reviewer #2:

Reviewer #2: The study by Wang et al. presents a new therapeutic platform to deliver thrombolysis after ischemic stroke in mice with diminished risk of hemorrhagic transformation (HT). The intervention is based on synthetic particles that are targeted to the thrombus by a fibrin-binding peptide and contain encapsulated rtPA. Furthermore, the authors loaded the particles with recombinant CD177 to block neutrophil-endothelial cell interaction which resulted in reduced BBB breakdown and neutrophil migration to the stroked brain. Combinatorial nanoparticles incorporating tPA and rCD177 proved more potent in preventing HT. This is a well-designed study that shows the efficacy of a novel ischemic stroke therapy with potential for clinical translation. However, the quantitative aspects of the study are limited, data are not well presented, and the benefits for stroke outcome have not been investigated. Specific concerns are:

Re: We sincerely thank the Reviewer for the thorough and positive assessment of our study, as well as for the constructive feedback aimed at further improving our manuscript. We have carefully considered each of the points raised and have revised the manuscript accordingly to strengthen the quantitative analysis, data presentation, and investigation of functional outcomes. Below we provide a point-by-point response to the specific concerns mentioned.

1. There is no indication that CP@tPA reduced infarct volume and improved neurobehavioral outcome after stroke. These are essential – and the only clinically

relevant – parameters for any therapeutic intervention.

Re: Thanks very much for your valuable suggestion, which aligns with our understanding that assessing both infarct volume and functional neurological outcomes is crucial for evaluating the therapeutic potential of any stroke treatment. In response to your comment, we have conducted additional triphenyltetrazolium chloride (TTC) staining to specifically assess the effect of nanodrug therapy on reducing brain infarct volume. As demonstrated in **Supplementary Fig. 29**, MCAO mice developed substantial cerebral infarction, exhibiting an infarct volume of 35.68% at 24 h poststroke. Administration of free tPA and CP@tPA reduced the infarct volume to 28.07% and 19.91%, respectively. Notably, preintervention with CP@rCD177 resulted in a further 39.21% reduction in infarct volume compared to CP@tPA group, culminating in a final infarct volume of only 12.1%. This outcome underscores its superior efficacy in mitigating ischemic brain damage by inhibiting CD177⁺ neutrophil infiltration.

Supplementary Fig. 29. a, TTC staining and **b**, infarct ratio of MCAO mice at 24 h poststroke ($n = 6$). Data are presented as the mean \pm s.d. P values were calculated by one-way ANOVA followed by Tukey's post-hoc test.

We next assessed the long-term neurobehavioral recovery in mice through a comprehensive behavioral test, which incorporated multiple sensorimotor assessments including the mNSS, corner turn, rotarod, and adhesive removal tests, as well as the

Barnes maze test to evaluate cognitive function. Our results revealed that both MCAO surgery and post-thrombolysis led to severe motor deficits, accompanied by significantly elevated mNSS scores (**Supplementary Fig. 32**). The functional impairments associated with tPA thrombolysis were closely correlated with the hemorrhagic complications it induced.¹⁻³ In contrast, pretreatment with CP@rCD177 prior to thrombolysis markedly improved functional outcomes: reduced mNSS scores, more symmetrical turning behavior, shorter removal time in the adhesive test, and longer latency to fall on the rotarod (**Supplementary Fig. 33a-g**). Spatial memory was assessed using the Barnes maze on day 28 post-treatment (**Supplementary Fig. 33h**). Mice in the MCAO, tPA, and CP@tPA groups required significantly more time, traveled longer distances, and made more errors before locating the escape box compared to sham mice (**Supplementary Fig. 33i-k**). CP@rCD177 treatment effectively attenuated these deficits, indicating preserved cognitive function. Together, these results demonstrate that the combine use of CP@rCD177 and CP@tPA not only reduces infarct size but also promotes significant recovery of sensorimotor and cognitive functions, highlighting its potential as a promising therapeutic strategy for ischemic stroke.

Supplementary Fig. 32. Neurological deficits were measured by assessing mNSS score ($n = 6$). Data are presented as the mean \pm s.d. P values were calculated by two-way ANOVA followed by Tukey's post-hoc test.

Supplementary Fig. 33. Long-term neurological recovery after stroke. **a**, Representative photo of the corner turn test. **b**, Laterality index measured in the corner turn test ($n = 6$). **c**, Representative photo of the rotarod test. **d**, Mean latency to fall during the rotarod test ($n = 6$). **e**, Schematic illustration of the adhesive test. **f**, Time to contact the tape in the adhesive test ($n = 6$). **g**, Time to remove the tape in the adhesive test ($n = 6$). **h**, Schematic of Barnes maze test. **i**, Number of errors and **j**) latency to find the escape hole in the Barnes maze test ($n = 6$). **k**, Representative images of walking path in different groups at day 28 after MCAO. Data are presented as the mean \pm s.d. P values were calculated by one-way or two-way ANOVA followed by Tukey's post-hoc test.

Reference:

1. Shi, K. et al. tPA mobilizes immune cells that exacerbate hemorrhagic transformation in stroke. *Circ. Res.* **128**, 62-75 (2021).
2. Mao, L. et al. Regulatory T cells ameliorate tissue plasminogen activator-induced brain haemorrhage after stroke. *Brain.* **140**, 1914-1931 (2017).
3. Shi, SX. et al. CD4⁺ T cells aggravate hemorrhagic brain injury, *Sci. Adv.* **9**, eabq0712 (2023).

2. *It is presumed that 100 μM H_2O_2 was used for all in vitro experiments. The concentrations of extracellular H_2O_2 released by neutrophils is thought to be in the low μM range (PMID: 6323407). In addition, the presence of erythrocytes in the clot would act as a H_2O_2 sponge further reducing available H_2O_2 for polymer breakdown.*

Re: Thank you for your insightful comment regarding the concentration of H_2O_2 used in our in vitro experiments. We fully appreciate your consideration of physiological H_2O_2 levels derived from neutrophils and the potential scavenging effect of erythrocytes. We would like to clarify that our drug release mechanism is designed to respond not to neutrophil-derived ROS, but to the intrinsically elevated levels of ROS within the thrombus microenvironment. Vascular occlusion by a thrombus induces localized hypoxia and endothelial injury, which significantly promote reactive oxygen species generation.¹ Specifically, vascular-derived H_2O_2 serves as a key mediator of platelet activation and aggregation, while the activated platelets themselves further produce substantial H_2O_2 , creating a pathogenic cascade that amplifies platelet recruitment, enhances thrombus growth, and accelerates inflammatory progression.² This self-amplifying process leads to markedly elevated H_2O_2 concentrations within the thrombus, ranging from micromolar to millimolar levels ($>10 \mu\text{M}$), with millimolar concentrations being attained near the injured endothelium.^{3,4} Moreover, the pathological oxidative environment can convert Fe^{2+} -containing hemoglobin to Fe^{3+} methemoglobin, further exacerbating oxidative stress and promoting thrombosis.⁵ Therefore, the use of $100 \mu\text{M}$ H_2O_2 in our in vitro system represents a well-established approach to simulate such pathological oxidative stress conditions, as this concentration aligns with reported physiological ranges in thrombosis and has been widely adopted in thrombolysis research.^{6,7}

Reference:

1. Quan, X. et al. Cryo-shocked platelet coupled with ros-responsive nanomedicine for targeted treatment of thromboembolic disease. *ACS Nano*. **17**, 6519-6533 (2023).
2. Kang, C. et al. Fibrin-targeted and H_2O_2 -responsive nanoparticles as a theranostics for

- thrombosed vessels. *ACS Nano*. **11**, 6194-6203 (2017).
3. Zhang, H. et al. On-site self-penetrating nanomedicine enabling dual-priming drug activation and inside-out thrombus ablation. *ACS nano*. **18**, 34683–34697 (2024).
 4. Sies, H. et al. Reactive oxygen species (ROS) as pleiotropic physiological signalling agents. **21**, 363–383 (2020).
 5. Gutmann, C. et al. Reactive Oxygen Species in Venous Thrombosis. *Int J Mol Sci*. **21**, 1918 (2020).
 6. Lu, Y. et al. Microthrombus-targeting micelles for neurovascular remodeling and enhanced microcirculatory perfusion in acute ischemic stroke. *Adv. Mater.* **31**, e1808361 (2019).
 7. Zhao Y. et al. Biomimetic fibrin-targeted and H₂O₂-responsive nanocarriers for thrombus therapy. *Nano Today* **35**, 100986 (2020).

3. Ferric chloride clots are relative resistant to tPA thrombolysis because of low fibrin incorporation into the clot (PMID: 21326267). What was the recanalization rate in tPA and CP@tPA treated FeCl₃ MCAO mice? Why would CP@tPA be superior in dissolving a clot that has minimal fibrin deposition?

Re: Thank you for raising this important point regarding the fibrin content and thrombolytic susceptibility of FeCl₃-induced thrombi. We appreciate the opportunity to clarify the mechanism and efficacy of CP@tPA in this context. To test this, we established FeCl₃-induced MCAO model, treatment with CP@tPA at 2 h post-occlusion resulted in significant cerebral blood flow restoration, reaching 65.3 ± 5.38% as measured by laser speckle imaging (**Fig. R3**). We acknowledge that FeCl₃-induced thrombi are characterized by a relatively lower overall fibrin content compared to other thrombus types. However, as noted in the literature,¹ fibrin, though less abundant, still plays an essential structural role in maintaining clot stability. Importantly, the fibrin network in such thrombi tends to be looser and less densely cross-linked. This structural feature, rather than being a barrier to thrombolysis, may render the clot more susceptible to targeted degradation. The more open architecture facilitates enhanced penetration and accessibility for fibrinolytic agents.

CP@tPA is specifically designed to home to the thrombus site and release tPA in a localized and sustained manner. By concentrating its activity precisely at sites where the fibrin matrix is critically located (even if sparse), CP@tPA efficiently dismantles this key structural component, thereby achieving targeted therapeutic impact. This approach overcomes the limitation of low overall fibrin density by maximizing the local enzymatic impact on the vulnerable, loosely organized network. Thus, the recanalization achieved with CP@tPA in our model can be attributed to its ability to leverage the structural vulnerability of the fibrin component through site-specific drug delivery, leading to effective clot dissolution despite the lower fibrin incorporation characteristic of FeCl₃-induced thrombi.

Fig. R3. Laser speckle contrast images (a) and rCBF quantification (b) of FeCl₃-induced MCAO model in different treatment groups ($n = 4$). Data are presented as the mean \pm s.d. P values were calculated by one-way ANOVA or two-way ANOVA followed by Bonferroni's post-hoc test.

Reference:

1. Karatas, H. et al. Thrombotic distal middle cerebral artery occlusion produced by topical FeCl₃ application: a novel model suitable for intravital microscopy and thrombolysis studies. *J Cereb Blood Flow Metab.* 31, 1452-1460 (2011).

4. Fig 2h. The HT in the CP@tPA coronal sections looks different between tPA and CP@tPA treated animals. While the number of macroscopic petechial bleeds might be higher in tPA treated mice, there is indication of a more diffuse blood extravasation in

CP@tPA mice. At least this is my impression from the presented images. In addition, petechial bleeds are also observed in the contralateral hemisphere of CP@tPA mice. Contralateral and ipsilateral hemispheres of all treatment groups should be used to quantitatively determine the hemoglobin content by chemical analysis.

Re: We appreciate your insightful observations regarding the hemorrhage patterns in our study. And we have provided clarification below addressing the differential hemorrhage morphology and contralateral bleeding.

Regarding the difference in hemorrhage morphology:

You correctly noted that while the tPA group exhibited predominantly macroscopic petechial hemorrhages, the CP@tPA group showed a more diffuse pattern of blood extravasation (**Fig. 2h**). We hypothesize that this may be attributed to partial thrombus fragmentation during the continuous thrombolytic process induced by CP@tPA. These micro-thrombi may embolize to distal microvascular beds within the ischemic hemisphere, leading to multiple foci of secondary hemorrhage upon further lysis, thereby resulting in a more disseminated bleeding pattern.¹

Fig. 2h. Digital photos showing cerebral hemorrhage in MCAO mice.

Regarding contralateral hemorrhage:

We also observed punctate hemorrhages in the contralateral, non-ischemic hemisphere, particularly in groups with more severe ipsilateral HT. This phenomenon is likely related to a systemic inflammatory response triggered by the initial hemorrhagic transformation. Circulating inflammatory mediators may compromise vascular integrity in the contralateral hemisphere, leading to secondary micro-

hemorrhages.² In the absence of the primary ischemic insult, hemorrhage in the contralateral hemisphere remains mild and focal, predominantly manifesting as petechiae, and is consequently significantly less severe than that in the ipsilateral hemisphere.

Quantitative confirmation:

In line with your suggestion, we performed quantitative hemoglobin assays on both cerebral hemispheres. The results confirmed the presence of low but detectable levels of hemorrhage in the contralateral hemisphere of tPA- and CP@tPA-treated mice with significant ipsilateral HT (**Fig. R4**). This supports the histological observations and indicates that while contralateral hemorrhage is milder, it is indeed a real and quantifiable phenomenon.

We thank you again for raising these important points, which have allowed us to further clarify the spatial characteristics and systemic implications of hemorrhagic transformation in our model.

Fig. R4. Hemoglobin content in the contralateral and ipsilateral hemispheres of mice in different treatment groups ($n = 7$). Data are presented as the mean \pm s.d. P values were calculated by one-way ANOVA followed by Bonferroni’s post-hoc test.

Reference:

1. Jacqmarcq, C. et al. MRI-based microthrombi detection in stroke with polydopamine iron oxide. *Nat. Commun.* **15**, 5070 (2024).
2. Sun, S. et al. Smart liposomal nanocarrier enhanced the treatment of ischemic stroke through

neutrophil extracellular traps and cyclic guanosine monophosphate-adenosine monophosphate synthase-stimulator of interferon genes (cGAS-STING) pathway inhibition of ischemic penumbra. *ACS Nano*. **17**, 17845-17857 (2023).

5. Fig S9 shows extensive accumulation of polymersomes in lung, kidney, and liver. How and when are polymersomes cleared from those organs and what is their effect on lung, kidney, and liver tissue integrity and function?

Re: Thank you for raising this important question regarding the biodistribution and potential side effects of our polymersomes. At 2 h post-injection, CP polymersomes predominantly accumulated in the liver, kidneys, and lungs (**Supplementary Fig. 12**), which is consistent with typical clearance pathways for systemically administered nanoparticles.¹ Over time, the polymersomes can be cleared primarily via hepatic processing and biliary-fecal excretion.² To validate the in vivo clearance kinetics of the CP polymersomes, we performed in vivo fluorescence imaging at extended time points including 24 h, 48 h, 72 h, and one week. The results showed a significant decrease in fluorescence signals in these organs within 72 h, with a return to baseline levels within one week (**Fig. R5**), demonstrating complete metabolic clearance.

Fig. R5. a, *Ex vivo* fluorescence imaging of CP@ICG in mouse tissues at different time points post-injection. H, heart; Li, liver; S, spleen; Lu, lung; K, kidney. **b,** The average radiation efficiency of CP@ICG in mouse tissues based on fluorescence intensity ($n = 3$). Data are presented as the mean \pm s.d. P values were calculated by One-way ANOVA followed by Bonferroni's post-hoc test.

To assess whether this transient accumulation adversely affected tissue integrity or organ function, we performed comprehensive histopathological and biochemical analyses. H&E staining of major organs (heart, liver, spleen, lungs, and kidneys) revealed no signs of abnormalities, lesions, or inflammation in the CP@tPA-treated group compared with sham mice (**Supplementary Fig. 13a**). Furthermore, serum biochemical analysis showed no significant differences in key markers of hepatic (ALT, AST, ALP) and renal (CREA, UREA, UA) functions between the CP@tPA and sham groups (**Supplementary Fig. 13b**). Taken together, these data demonstrate that although CP polymersomes transiently accumulate in reticuloendothelial organs, they are effectively cleared within a week and do not elicit significant toxicological effects on tissue structure or function, supporting the biocompatibility and safety of CP@tPA for *in vivo* application.

Supplementary Fig. 13. H&E staining of heart, liver, spleen, lung, and kidney (**a**) and serum biochemical indices (**b**) in sham and CP@tPA groups 24 h poststroke ($n = 3$).

Data are presented as the mean \pm s.d. *P* values were calculated by two-tailed unpaired *t*-test.

Reference:

1. Zhang, A. et al. Absorption, distribution, metabolism, and excretion of nanocarriers in vivo and their influences. *Adv. Colloid Interface Sci.* **284**, 102261 (2020).
2. Xue, W. et al. Effects of core size and PEG coating layer of iron oxide nanoparticles on the distribution and metabolism in mice. *Int. J. Nanomedicine.* **13**, 5719-5731 (2018).

6. Many experiments are missing time points of drug treatment and endpoints. Generally, time points should be indicated in the main text and/or figure legends, and not in the methods section.

Re: Thank you for this constructive suggestion. We realize that clearly indicating the timing of interventions and experimental endpoints is critical for interpreting the results. As suggested, we have now clearly stated all relevant time points, including the timing of drug administration and endpoint assessments, directly in the main text and the figure legends of the revised manuscript and supporting information. This ensures that readers can readily access key temporal information without needing to refer to the Methods section. We believe these revisions significantly improve the clarity and reproducibility of our study.

7. The SYTOX indicator should be used in conjunction with a neutrophil marker such as Ly6G.

Re: We sincerely thank the reviewer for the insightful comment. Colocalization analysis between Ly6G and SYTOX Green signals can confirm that the detected NETs predominantly originated from neutrophils. As suggested, we have performed multiplex immunofluorescence staining to visualize neutrophil infiltration (Ly6G⁺), NET formation (SYTOX Green⁺), and vascular leakage (Evans Blue) in mouse brains 24 h

after treatment. Intravital microscopy revealed minimal neutrophil adhesion, extravasation, NET deposition, or Evans Blue leakage in sham-operated mice (**Fig. 5f**). In contrast, the HT group exhibited substantial neutrophil infiltration and extensive NET formation at sites of vascular leakage, supporting the notion that neutrophil-derived NETs contribute significantly to BBB disruption.¹ CP@rCD177 treatment markedly reduced neutrophil infiltration, suppressed NET deposition, and attenuated Evans Blue extravasation. These results indicate the utility of combining Ly6G and SYTOX Green staining to accurately attribute NET formation to neutrophils and to evaluate the therapeutic effect of CP@rCD177 on NET-associated BBB damage.

Fig. 5f, Representative immunofluorescence staining images showing neutrophil infiltration (Ly6G⁺, red), NET distribution (SYTOX⁺, blue) and vascular leakage (Evans blue, green) in mouse brains 22 h after therapy.

Reference:

1. Kang, L. et al. Neutrophil extracellular traps released by neutrophils impair revascularization and vascular remodeling after stroke. *Nat. Commun.* **11**, 2488 (2020).

Minor

1. In what volume were the polymersomes and tPA delivered?

Re: Thank you for raising this important point regarding the administration details. In

our study, both the polymersomes and free tPA were administered intravenously via the tail vein in a total volume of 150 μ L per mouse. The dose of tPA (either in free form or encapsulated within the polymersomes) was 1 mg/kg. This information has now been clearly stated in the Method section of the revised manuscript.

2. There are no clinical data for stroke patients and controls.

Re: We thank the reviewer for raising this important point regarding clinical controls. In response, we would like to clarify the rationale and design of our control groups. The primary aim of our study was to investigate mechanisms specifically linked to tPA-induced hemorrhagic transformation in acute ischemic stroke. For this purpose, we compared transcriptomic profiles from peripheral blood of stroke patients who received tPA and later developed HT (the HT group) against those before tPA thrombolysis (the stroke group). This comparison was intentionally designed to isolate molecular signatures associated with tPA-related HT, rather than stroke in general. That said, we recognize the value of broader contextualization. Therefore, we also included a normal control group consisting of healthy volunteers.

[editorial note: unpublished, confidential data redacted]

Importantly, during the HT phase, neutrophil activation and NET formation were

further amplified (**Fig. 3c,d; Supplementary Fig. 16 and Fig. 17**), reinforcing their potential role in driving HT progression. We fully agree that additional clinical control data could further strengthen the interpretability of our findings. We intend to incorporate such comparisons in subsequent clinical validation studies. We thank the reviewer again for this constructive suggestion, which has helped us better articulate the scope and implications of our work.

[editorial note: unpublished, confidential data redacted]

Reference:

1. Zhang, X. et al. Down-regulated cylindromatosis enhances NF- κ B activation and aggravates inflammation in HBV-ACLF patients. *Emerg. Microbes Infect.* **11**, 1586-1601 (2022).
2. Xu, H. et al. New genetic and epigenetic insights into the chemokine system: the latest discoveries aiding progression toward precision medicine. *Cell. Mol. Immunol.* **20**, 739–776

(2023).

3. Wang, Y. et al. Neutrophil extracellular trap-microparticle complexes trigger neutrophil recruitment via high-mobility group protein 1 (HMGB1)-toll-like receptors (TLR2)/TLR4 signalling. *Br. J. Pharmacol.* **176**, 3350-3363 (2019).
4. 3.Hamam, H. et al. Histone Acetylation Promotes Neutrophil Extracellular Trap Formation. *Biomolecules* **9**, 32 (2019).
5. 4.Muñoz-Caro, T. et al. The Role of TLR2 and TLR4 in Recognition and Uptake of the Apicomplexan Parasite *Eimeria bovis* and Their Effects on NET Formation. *Pathogens* **10**, 118 (2021).

3. *Methods for RNA-seq and DGE, including statistical analysis, are missing.*

Re: We thank the reviewer for pointing out this omission. We have supplemented a detailed description of the RNA-seq and DGE analysis methods in the Method Section of the revised manuscript. Specifically, the added content includes information on read quantification using TPM and RSEM, statistical analysis using DESeq2 with cut-offs of $|\log_2FC| \geq 1$ and $FDR \leq 0.05$, and functional enrichment analysis using GO, KEGG, and Reactome databases with Goatools and KOBAS tools. We sincerely appreciate the reviewer's feedback, which has helped improve the clarity and completeness of our methodology.

4. *Line 504. Please specify "lower extremity artery".*

Re: Thanks for pointing out the need for a more precise description of lower extremity arterial thrombus model. In the revised manuscript, we have provided a detailed protocol specifying that thrombus formation was induced in the femoral artery through topical application of 10% FeCl₃ for 1 min. The specific surgical steps for vessel exposure, the exact size of the filter paper (1 × 1 mm), and the method for confirming thrombus formation have also been clearly stated to enhance reproducibility.

5. Fig 1e-f. Reversed order in legend.

Re: We sincerely appreciate the reviewer for pointing out this error. We have corrected the reversed order in the legend of **Fig. 1e** and **1f** in the revised manuscript to accurately match the corresponding image panels. We appreciate the reviewer's careful reading and helpful comment, which has improved the clarity and accuracy of our figure presentation.

6. Fig S7a. Size markers missing.

Re: Thank you for highlighting this omission. We have now added clear size markers to **Supplementary Fig. 7a** (now moved to **Supplementary Fig. 10a**) to facilitate accurate interpretation of the results. We appreciate the reviewer's careful attention to detail, which has helped enhance the clarity and accuracy of our supplementary materials.

With Regard to the Comments of Reviewer #2:

Reviewer #2: The authors have added relevant clarifications and provided new data and experiments to support their findings, thereby improving the overall quality of the study. One concern remains, that needs to be addressed. While there is an overall decrease in petechiae and larger bleedings in CP@tPA treated animals, the total amount of brain hemoglobin was not reduced (Fig. R4). Furthermore, the authors refer to Fig. 4J as quantitative hemoglobin measurements (“hemoglobin detection kit”) while the graph seems to depict hemoglobin volume obtained by morphometric analysis. This discrepancy needs to be addressed.

In addition, the presence of diffuse hemoglobin signals in CP@tPA might reflect microvascular problems, as acknowledged by the authors in their rebuttal, and might be an important factor for the clinical translation of this therapeutic approach. It needs, therefore, further investigation (high resolution histology?), and Fig R4 should be presented as supplemental data. The authors are also encouraged to include these findings into the discussion.

Re: Thank you for your insightful comments and for acknowledging the improvements made to our manuscript. We appreciate the opportunity to address the remaining concern regarding the relationship between macroscopic bleeding reduction and total brain hemoglobin content, as well as the clarification on our quantification method. We agree that this point is critical for the interpretation of our findings and its clinical implications. Regarding the methodological clarification for **Fig. 4J**, we apologize for any confusion caused by the initial wording. The quantitative data presented were obtained not through morphometric analysis, but by homogenizing whole-brain tissue and using a hemoglobin assay kit. The values were calibrated against a standard curve generated from the blood of the same mouse strain and are expressed as hemorrhage volume.² We have revised the Methods section to describe this procedure with greater precision and clarity.

We fully acknowledge the central point of your critique: the observation that while CP@tPA treatment reduces visible petechiae and larger hemorrhages, it does not decrease the total brain hemoglobin content (now presented in **Supplementary Figure 16**). We believe this apparent discrepancy is highly informative. It aligns with our data in **Fig. 2h**, confirming that the enhanced thrombolytic efficiency of CP@tPA is indeed accompanied by a more widespread hemorrhagic transformation. This suggests a shift in the bleeding pattern, where focal bleeding may be mitigated but is supplanted by a more diffuse, distributed extravasation of blood throughout the parenchyma. This phenomenon directly supports your hypothesis of underlying "microvascular problems," indicating that the therapeutic intervention, while effective in lysing the primary clot, may inadvertently exacerbate capillary fragility and leakage. This critical limitation underscores why bleeding complications remain a severe constraint for clinical translation¹ and powerfully validate the necessity of our follow-up strategy involving rCD177 as an adjunctive therapy designed precisely to protect vascular integrity.

Supplementary Fig. 16. Hemoglobin content in the contralateral and ipsilateral hemispheres of mice in different treatment groups ($n = 7$). Data are presented as the mean \pm s.d. P values were calculated by one-way ANOVA followed by Bonferroni's post-hoc test.

In direct response to your suggestion for further investigation, we performed additional high-resolution histological and functional assessments of cerebral microvasculature. In vivo intravital microscopy³ revealed that CP@tPA treatment led to a more extensive distribution of Evans blue leakage sites compared to tPA alone (**Supplementary Fig. 30**), confirming a pattern of diffuse vascular permeability. Importantly, adjunctive treatment with CP@rCD177 significantly reduced this extravasation. Furthermore, ultrastructural analysis by TEM demonstrated that the blood-brain barrier microstructure remained compromised after CP@tPA treatment, evidenced by persistent inter-endothelial gaps and loss of tight junctions (**Supplementary Fig. 31**). CP@tPA did not reduce the number of these pathological gaps compared to tPA alone, providing a structural basis for the diffuse hemorrhage. In contrast, CP@rCD177 administration effectively reversed this damage, reducing the number of inter-endothelial gaps to approximately 49% of those in the CP@tPA group. These findings substantiate that the diffuse hemoglobin signal reflects authentic microvascular injury and demonstrate that our combinatorial approach with rCD177 specifically targets and ameliorates this pathology.

Supplementary Fig. 30. Intravital microscopy images of intravenously injected Evans' blue leakage in the cortical vessels at 22 h post-treatment.

Supplementary Fig. 31. a, TEM images illustrate the ultrastructure of the BBB in each treatment group. Green arrow, intact tight junction; purple arrow, open tight junction; red arrow, inter-endothelial gap. **b**, Number of gaps per 1 mm vessel perimeter in each treatment group ($n = 6$). Data are presented as the mean \pm s.d. P values were calculated by one-way ANOVA followed by Tukey's post-hoc test.

We have integrated these insights into the revised manuscript. As advised, the former **Fig. R4** is now included as **Supplementary Fig. 16**, and the discussion section has been expanded to explicitly address the nature of diffuse microbleeds as a translational challenge, and the mechanistic role of CP@rCD177 in stabilizing the blood-brain barrier. We are grateful for your expert guidance, which has prompted these essential analyses and significantly strengthened the mechanistic narrative of our study. We hope our detailed response and the corresponding revisions in the manuscript fully address your concerns.

Reference:

1. Wang, R. et al. Neutrophil extracellular traps promote tPA-induced brain hemorrhage via cGAS in mice with stroke. *Blood*, **138**, 91–103 (2021).
2. Burk, J. et al. Protection of cerebral microvasculature after moderate hypothermia following experimental focal cerebral ischemia in mice. *Brain Res.* **1226**, 248–255 (2008).
3. Kang, L. et al. Neutrophil extracellular traps released by neutrophils impair revascularization and vascular remodeling after stroke. *Nat. Commun.* **11**, 2488 (2020).